Ontogenetic braincase development in Psittacosaurus lujiatunensis (Dinosauria: Ceratopsia) using micro-computed tomography

Bullar Claire M. 1 cb14233@bristol.ac.uk
Zhao Qi 1 2
http://orcid.org/0000-0002-4323-1824 Benton Michael J. 1
Ryan Michael J. 3
1 School of Earth Sciences, University of Bristol , Bristol , UK
2 Institute of Vertebrate Paleontology and Paleoanthropology, Chinese Academy of Sciences , Beijing , China
3 Department of Earth Sciences, Carleton University , Ottawa, ON , Canada
Knoll Fabien
Electronic publication date: 2019 Aug 14
Publication date: 2019
Volume: 7
Electronic Location ID: e7217
Received 2018 Oct 23; Accepted 2019 May 31
Copyright: © 2019 Bullar et al.
Copyright year: 2019
Copyright holder: Bullar et al.
License: This is an open access article distributed under the terms of the Creative Commons Attribution License, which permits unrestricted use, distribution, reproduction and adaptation in any medium and for any purpose provided that it is properly attributed. For attribution, the original author(s), title, publication source (PeerJ) and either DOI or URL of the article must be cited.
License URL: https://creativecommons.org/licenses/by/4.0/

Keywords: Dinosaurs, Braincase, Ceratopsian, Ontogeny, Morphology, Tomography, Psittacosaurus

Funding: Geological Society of London (William George Fearnsides Fund) and by NERC Standard Grant NE/I027630/1 Newton Advanced Fellowships of the Royal Society NA160290 Strategic Priority Research Program of Chinese Academy of Sciences XDB26000000, XDB183030504 This work was supported by the Geological Society of London (William George Fearnsides Fund) and by NERC Standard Grant NE/I027630/1 awarded to Michael J Benton. It was also supported by the Newton Advanced Fellowships of the Royal Society (NA160290) and the Strategic Priority Research Program of Chinese Academy of Sciences (XDB26000000, XDB183030504) awarded to Qi Zhao. The funders had no role in study design, data collection and analysis, decision to publish, or preparation of the manuscript.

==============================
Ontogenetic sequences are relatively rare among dinosaurs, with Ceratopsia being one of the better represented clades, and especially among geologically earlier forms, such as Psittacosaurus. Psittacosaurus is a small, bipedal basal ceratopsian abundant in the Lower Cretaceous deposits of Asia, whose cranial and endocranial morphology has been well studied, but only cursory details have been published on the bones surrounding the brain. Using reconstructions created from micro-computed tomography scans of well-preserved skulls from the Barremian–Aptian Yixian Formation, China, we document morphological changes in the braincase of Psittacosaurus lujiatunensis through three growth stages, hatchling, juvenile, and adult, thus providing the first detailed study of ceratopsian braincase morphology through ontogeny. Notable ontogenetic changes in the braincase of P. lujiatunensis include a dramatic relative reduction in size of the supraoccipital, an increase in the lateral expansion of the paroccipital processes and a decrease in the angle between the lateral semicircular canal and the palatal plane. These ontogenetic morphological changes in the braincase relate to expansion of the cranium and brain through growth, as well as reflecting the switch from quadrupedal juveniles to bipedal adults as documented in the changing orientation of the horizontal semicircular canal through ontogeny. Recognition of these patterns in a basal ceratopsian has implications for understanding key events in later ceratopsian evolution, such as the development of the parieto-squamosal frill in derived neoceratopsians.

Introduction

Psittacosaurus is an Early Cretaceous genus of Ceratopsia, a diverse and geographically widespread suborder of ornithischian dinosaurs (Dodson, Forster & Sampson, 2004; Chinnery-Allgeier & Kirkland, 2010; Dodson, 2013). Their position within ceratopsian phylogeny has been debated, from being basalmost (Sereno, 2000; Morschhauser, 2012; Han et al., 2017), to more derived than Yinlong and chaoyangsaurids (Xu et al., 2006; He et al., 2015; Zheng, Jin & Xu, 2015). Recent work by Han et al. (2015, 2016, 2017) has suggested a close relationship with chaoyangsaurids. Chaoyangsauridae is a family of basal ceratopsian dinosaurs, that is either sister to Neoceratopsia (Xu et al., 2002; Han et al., 2017) or its most basal member (Sereno, 2000; You & Dodson, 2003).

Ceratopsia underwent a shift from bipedalism to quadrupedalism during their evolution (Weishampel et al., 2004; Xu et al., 2006). Conversely, Psittacosaurus underwent a significant postural shift during ontogeny, as quadrupedal hatchlings developed into facultative bipeds during growth. Evidence for the postural shift comes from allometric studies of limb ratios and histology (Zhao et al., 2013, 2014). Skeletal evidence shows relative shortening of the forelimbs during growth, and histological evidence shows relative slowing of growth of the forelimbs and increasing growth rates for hindlimb elements (femur, tibia) when they reached maturity, at an age of approximately 4 years (Zhao et al., 2013). P. mongoliensis, P. sibiricus and P. lujiatunensis are the largest species of Psittacosaurus, all three reaching up to two m in length. P. lujiatunensis is prevalent in the Early Cretaceous Lujiatun deposits of the Yixian Formation in Liaoning Province of China. Hedrick & Dodson (2013) posited that three species of Lujiatun Psittacosaurus (Hongshanosaurus houi, Psittacosaurus lujiatunensis and P. major) are synonymous and represent different taphomorphotypes rather than individual species. Here, we accept their results and follow their taxonomic assignments. Associated volcanic beds have provided radioisotopic dates for the Lujiatun volcaniclastic sediments of approximately 126 Ma (Chang et al., 2017). The fossiliferous layers exhibit evidence of rapid burial due to volcanic activity (Rogers et al., 2015) and subsequent exceptional preservation.

The integration of advanced computed tomography (CT) techniques into paleontological studies has enabled extensive, in-depth morphological reviews and non-destructive investigations into both extinct and extant taxa (Chapelle & Choiniere, 2018; Hoffman, Heckert & Zanno, 2018; Neenan & Scheyer, 2012; Walsh & Knoll, 2018). While several studies have touched upon the braincase of Psittacosaurus (Coombs, 1982; Zhou et al., 2007; Dodson, You & Tanoue, 2010; Sereno, 2010), few have used CT data to explore internal structures (Zhou et al., 2007).

Here, we describe three braincases, each belonging to individuals of P. lujiatunensis at different ontogenetic stages. Using the methodology of Zhao et al. (2014), we determined the approximate ages of these specimens from their base skull lengths. Using sutural fusion as a proxy for age is ambiguous at best and varies in different taxa and skeletal location (Brown & Schlaikjer, 1940; Scannella & Horner, 2011; Longrich & Field, 2012; Bailleul et al., 2016). When used in conjunction with other techniques, however, sutural fusion has been accepted as a rough proxy for determining age in some vertebrates (Brochu, 1996; Sampson, Ryan & Tanke, 1997), and we follow this principle of using combined evidence.

Materials and Methods

The smallest skull (IVPP V15451) belongs to a hatchling under 1 year old, as determined by its size, lack of fusion, and partial disarticulation within the braincase (Fig. 1). The second specimen (IVPP V22647) is fully articulated and well fused (Fig. 1). The level of fusion present and the size of the specimen indicates an age of approximately 2 years. The largest skull (IVPP V12617) is a fully fused mature Psittacosaurus, determined to be 10 years old at the time of death, based on the lines of arrested growth (Zhao et al., 2013) (Fig. 1).

Figure 1 Ontogenetic sequence of P. lujiatunensis.

(A) Hatchling (IVPP V15451) in lateral view. (B) Hatchling in dorsal view. (C) Juvenile (IVPP V22647) in lateral view. (D) Juvenile in dorsal view. (E) Adult (IVPP V12617) in lateral view. (F) Adult in dorsal view. All shown to the same scale; scale bar represents 20 mm.

Specimens were scanned using the Chinese Academy of Sciences micro-computed tomography scanner at the Institute of Vertebrate Palaeontology and Palaeoanthropology (IVPP), Beijing. Avizo 8 (FEI Visualization Sciences Group, Hillsboro, OR, USA) was used to virtually reconstruct and segment the braincase into 3D models. The hatchling skull was partially disarticulated, and so segmentation was relatively quick and straightforward, but both older specimens were well fused in comparison. Using structures such as semicircular canals, cranial nerve pathways, arteries, and undulations on the bone surface, we were able to divide both braincases into their constituent elements. Measurements of individual bones were taken in Avizo and the angles between the plane of the lateral semicircular canal and the palatal plane were calculated following the method in Schellhorn (2018).

Results

Hatchling braincase description

The relatively undeformed skull of the hatchling Psittacosaurus (IVPP V15451) is compact and rounded in comparison to the older specimens (Fig. 1). It is complete except for the nasals, prefrontals and orbitosphenoids. It measures 23.6 mm long (rostral to occipital condyle (OC)) by 15.5 mm tall (basipterygoid processes to frontal-parietal contact).

The foramen magnum (FM) is circular (Fig. 2D). Approximately 70% of the wall of the FM is made up by the exoccipitals (lateral walls), the rest is an equal measure of supraoccipital dorsally and basioccipital (BO) ventrally.

Figure 2 Segmented braincase of a hatchling P. lujiatunensis (IVPP V15451).

(A) Lateral view. (B) Ventral view. (C) Dorsal view. (D) Posterior view. (E) Anterior view. bo, basioccipital; bpc, basisphenoid-parasphenoid complex; fr, frontal; ls, laterosphenoid; pa, parietal; pp, paroccipital processes; pr, prootic; so, supraoccipital. Scale bar represents 10 mm.

The basioccipital contacts the basisphenoid anteriorly and the exoccipitals laterodorsally (Fig. 3D). It measures 5.4 mm anteroposteriorly by 5.8 mm transversely across its widest point. It forms the posterior floor of the braincase and contributes to the majority of the basal tubera (Fig. 2A: bo) with the basisphenoid contributing to the anteroventral-most regions. The basal tubera hang 1.2 mm below the OC and are separated by a pronounced medial groove (Figs. 3C, 3E and 3F). Their apices are rounded, blunt and the orientation of the long axis of the tubera is in the sagittal plane (Fig. 3C: bt).

Figure 3 Basioccipital of hatchling P. lujiatunensis (IVPP V15451).

(A) Left lateral view. (B) Right lateral view. (C) Ventral view. (D) Dorsal view. (E) Posterior view. (F) Anterior view. boc, basioccipital condyle; bsas, basisphenoid articular surface; bt, basal tubera; btg, basal tubera groove; cdn, condylar neck; eoas, exoccipital articular surface; ng, neural groove. Scale bar represents 5 mm.

The OC is anteroposteriorly compressed resulting in an oval appearance in dorsal and ventral views (Fig. 3C), with a shallow dorsal depression for the entry of the spinal cord (Fig. 3E). It measures 3 mm wide by 2 mm high. The basioccipital makes up the entirety of the OC with the exoccipitals resting on either side of the FM depression (Fig. 2D: eoas). There is no dorsal restriction of the condylar neck and little lateral constriction. The condylar neck is ventrally constricted, with its maximum constriction occurring proximal to the condyle making it dorsoventrally compressed and relatively thin in lateral view (Figs. 2B and 3A).

The flat dorsal surface of the basioccipital is anteriorly inclined and slopes away from the notch formed by its contribution to the FM. Two ridges extend anterolaterally from the opening of the FM to a plateau that lies on the dorsal surface of the basal tubera. The exoccipital articular surfaces are lateral to these ridges (Fig. 3D: eoas). The basioccipital of the hatchling specimen only contributes to approximately 15% of the FM, but comprises the entire OC (Fig. 2D).

The basisphenoid contacts the prootic dorsally and the basioccipital posteriorly, but is indistinguishable from the parasphenoid even at this early stage of ontogeny. This basisphenoid-parasphenoid complex (BPC) is complete and well preserved comprising the anterior floor of the braincase (Fig. 2B: bs). It measures 10.3 mm long (without the contribution to the basal tubera). The most prominent features of the complex are the four processes that radiate from the corners of the cuboid basisphenoid body, the basal tubera and the basipterygoid processes. The basipterygoid processes project anterolaterally, diverge at an angle of approximately 81° and measure approximately 3 mm in length. The anterior tips of the basipterygoid processes are flared and flattened to articulate with the pterygoids (Fig. 4F: ptas). The ventral surface of the basisphenoid is concave. The troughed cultriform process of the parasphenoid extends anteriorly from the basisphenoid, passing beyond the distal-most surface of the basipterygoid processes (Fig. 4: cp, bpp). It measures 7 mm, approximately 2/3 of the total length of the parabasisphenoid. Note that for the purposes of these descriptions, the “troughs” that separate the cultriform process from the basipterygoid processes will be called the paracultriform troughs (Fig. 4D: pct). The paracultriform troughs in the juvenile specimen are gentle and rounded. The furrow that separates the basal tubera continues anteriorly and separates the basipterygoid processes.

Figure 4 Basisphenoid of hatchling P. lujiatunensis (IVPP V15451).

(A) Left lateral view. (B) Right lateral view. (C) Ventral view. (D) Dorsal view. (E) Posterior view. (F) Anterior view. boas, basioccipital articular surface; bpp, basipterygoid process; bt, basal tubera; cfo, carotid foramen; cp, cultriform process; pct, paracultriform trough; pras, prootic articular surface; ptas, pterygoid articular surface; st, sella turcica. Scale bar represents 5 mm.

The sella turcica (pituitary fossa) is a deep depression in the anterodorsal surface of the basisphenoid which would have held the pituitary gland (Fig. 4D: st). It is triangular when viewed dorsally. This fossa continues to the cerebral carotid artery canal which then divides into two circular carotid foramen which exit the basisphenoid laterally, ventral to the prootics. The ridges on either side of the trough of the cultriform process meet anterior to the sella turcica which is separated from the former by a large laterally compressed midline blade extending down the cultriform process. The basisphenoid meets the basioccipital posterodorsally at the basal tubera (Fig. 2B). The contribution of the basisphenoid to the basal tubera consists of thin, posteriorly projecting, plate-like extensions ventral to the main bulk of the tubera (approximately 20% of the tubera). A well pronounced, smooth, flattened ridge dominates the posterodorsal surface of the basisphenoid and forms the contact surface for the prootic (Figs. 4A, 4B, 4D and 4E: pras). This slightly anteriorly inclined platform extends laterally and anteriorly bordering the posterolateral edges of the sella turcica.

The sub-oval, plate-like supraoccipital dominates the occipital surface and tapers dorsally in thickness (Fig. 2D: so). It measures 8.7 mm wide by 3.6 mm tall. The dorsal margin of the supraoccipital articulates with the posterior most edge of the parietals, whilst the ventral border contacts the paroccipital processes. The boundaries with the paroccipital processes are poorly defined; however, the thin (<1 mm) supraoccipital flares lateroventrally to 2.4 mm to meet them (Fig. 5C: eoas). The supraoccipital contributes approximately 20% of the dorsal wall of the FM. In caudal view, a V-shaped notch approximately one-fifth the depth of the supraoccipital is present at the mid-point of the dorsal margin. (Fig. 5E). A low midline ridge extends from the base of the notch to the ventral margin (Fig. 5E: somr). The anterior surface is concave where the hindbrain would have sat (Figs. 5C, 5D and 5F). The posterior semicircular canal traverses through the lateral wings of the supraoccipital (Fig. 5F: sccp).

Figure 5 Supraoccipital of hatchling P. lujiatunensis (IVPP V15451).

(A) Left lateral view. (B) Right lateral view. (C) Ventral view. (D) Dorsal view. (E) Posterior view. (F) Anterior view. eoas, exoccipital articular surface; paas, parietal articular surface; pras, prootic articular surface; sccp, semicircular canal pathway; somr, supraoccipital midline ridge. Scale bar represents 5 mm.

The paroccipital processes make up most of the lateroposterior surface of the braincase (Fig. 2D: pp) and are composed of exoccipitals and opisthotics which are so well fused that their suture is obliterated. The processes contact the supraoccipital dorsally, the prootic anteriorly and the basioccipital ventrally; there is also a small area of contact with the parietal lateral to the supraoccipital contact. The anterior boundary of the paroccipital processes can only be inferred due to high levels of fusion in this region. The paroccipital processes of the hatchling specimen are stunted and subrectangular. A single paroccipital process has a length of 6.2 mm and a height of 2.8 mm giving a height/length ratio of almost 1:2. Cranial nerve pathways X–XII pass through the exoccipitals, exiting posterolaterally underneath the processes. They are visible when the braincase is viewed posteriorly. The exoccipital-opisthotic complex of the hatchling does not contribute to the OC, but does contribute approximately 70% of the wall of the FM (Figs. 2D and 6E). A trunk of the exoccipital extends medioventrally from the FM to contact the basioccipital at the top of the OC (Fig. 6D). The exoccipitals house a section of the semicircular canals. Dorsal to the paroccipital processes are unpaired foramina which most likely allowed for the passage of the vena capitis dorsalis (Fig. 6D: vcd?).

Figure 6 Paroccipital processes of hatchling P. lujiatunensis (IVPP V15451).

(A) Left lateral view. (B) Ventral view. (C) Dorsal view. (D) Posterior view. (E) Anterior view. boas, basioccipital articular surface; fm, foramen magnum; pop, paroccipital process; pras, prootic articular surface; sccp, semicircular canal pathway; soas, supraoccipital articular surface; vcd, vena capitis dorsalis; CN X, XI, exit for the vagus nerve and the accessory nerve respectively; CN XII, exit for the hypoglossal nerve. scale bar represents 5 mm.

The laterosphenoid articulates with the prootic posteriorly, the parietal posterodorsally and the frontal anterodorsally. It measures 5.2 mm by 4.2 mm and is blocky and triangular in lateral view. It has a flared “wishbone” shape in cross section (Fig. 7C). The laterosphenoid sits anterodorsally to the prootic and contributes to the anterolateral wall of the braincase (Fig. 2A: ls). The laterosphenoid at this stage in growth appears very fragile and thin. The medial surface of the laterosphenoid is concave and flares posteriorly resulting in a triangular appearance. The lateral surface of the laterosphenoid is convex and comes to a central point called the laterosphenoid head (Figs. 7A and 7B: lsh). The laterosphenoid makes up the dorsal boundary of the foramina for cranial nerve V (trigeminal nerve).

Figure 7 Laterosphenoid of hatchling P. lujiatunensis (IVPP V15451).

(A) Left lateral view. (B) Right lateral view. (C) Ventral view. (D) Dorsal view. (E) Posterior view. (F) Anterior view. fras, frontal articular surface; lsh, laterosphenoid head; pras, prootic articular surface; CN V, dorsal margin of trigeminal nerve. Scale bar represents 5 mm.

The prootic of Psittacosaurus is a complex element that forms much of the lateral braincase wall and houses the majority of the inner ear. The prootic contacts the paroccipital processes posteriorly, the laterosphenoid anteriorly, and the basisphenoid ventrally (Fig. 2A: pr). It measures 7.1 mm dorsoventrally and 3.9 mm mediolaterally. It is sub-triangular when viewed dorsally. It is morphologically similar to the prootic of the perinatal Alligator mississippiensis reconstructed in Dufeau & Witmer (2015). A large notch for cranial nerve V can be seen on the anterior surface of the prootic, but it is not entirely enclosed by the prootic (Figs. 8A–8D: cranial nerve, CN V). A prominent median ridge extends down the medial surface of the prootic (Fig. 8D: prmr). Anterior to this ridge lies a concavity spanning the entire height of the prootic. Within this anterior concavity and posterior to the trigeminal foramen lies the foramen for CN VII (facial nerve) (Figs. 8A–8D). Posterior to this median ridge lies the space for the cochlear duct and vestibule of the inner ear. The pathway of the anterior semicircular canal can be seen extending anteroposteriorly across the dorsal edge of the prootic when viewed dorsally (Fig. 8F: sccp). This section of the canals is loosely held within the prootic and is not entirely enclosed within the bone.

Figure 8 Prootic of hatchling P. lujiatunensis (IVPP V15451).

(A) Left lateral view. (B) Right lateral view. (C) Left medial view. (D) Right medial view. (E) Ventral view. (F) Dorsal view. (G) Posterior view. (H) Anterior view. bsas, basisphenoid articular surface; eoas, exoccipital articular surface; lsas, laterosphenoid articular surface; prmr, prootic midline ridge; sccp, semicircular canal pathway; CN V, trigeminal nerve; CN VII, facial nerve. Scale bar represents 5 mm.

The parietal is contacted by the frontal anteriorly, the laterosphenoid anteroventrally, the supraoccipital posteroventrally and the squamosals posterolaterally. It is located on the posterodorsal surface of the braincase and comprises approximately half of the dorsal surface of the braincase (Fig. 2C: pa). It measures 7.4 mm anteroposteriorly and 14.1 mm across its widest point. The posterior edge of the parietal is an inverted “V” shape in cross section, whilst the anterior is convex (Fig. 9). The ventral surface of the parietal is deeply concave which is mirrored in the convex dorsal surface continuing the rounded dome-like skull roof that originates from the frontal. The hatchling parietal lacks a sagittal ridge. The upper temporal fenestra forms rounded indentations in the lateral margins of the parietal and a process extends posteroventrally to contact the squamosals.

Figure 9 Parietal of hatchling P. lujiatunensis (IVPP V15451).

(A) Left lateral view. (B) Right lateral view. (C) Ventral view. (D) Dorsal view. (E) Posterior view. (F) Anterior view. fras, frontal articular surface; lsas, laterosphenoid articular surface; soas, supraoccipital articular surface; sqas, squamosal articular surface; UTF, upper temporal fenestra. Scale bar represents 5 mm.

Frontal contacts include the parietal posteriorly, the nasal anteriorly, the laterosphenoid posteroventrally and the postorbital posterolaterally (Fig. 10: paas, nas, lsas, poas). The prefrontals would also have contacted the frontal, but there is no indication of where this contact lies. The frontals measure 11.2 mm anteroposteriorly and 15.8 mm transversely across their widest point. The frontals of the hatchling exhibit extreme dorsal doming which contributes greatly to the overall roundness of the skull (Fig. 2). Two deep rounded concavities are present on the ventral side of these elements, which is the location of the cerebral hemispheres (Fig. 10C: cc). They measure 11.1 mm across their widest point. The hourglass indentation that marks the anterior regions of the brain is anterior to these bulbous depressions. The interfrontal suture cannot be discerned and there is no crest present on the dorsal surface of the frontal. The frontal is anteriorly transversely thin and widens posteriorly, the widest point being posterior to the orbits. Dorsally, two small (0.6 mm) epiossifications lie symmetrically either side of the midline—this is the first time these structures have been observed in Ceratopsia. The frontal contributes to the dorsal edge of the large eye socket. The eye socket creates a rounded concave border on the anterolateral edges of the frontal where a ventral lip is present (Figs. 10A and 10B: om). The supra-orbital wall is orientated in a sagittal plane, meaning that the wall is fully visible laterally. In dorsal and ventral views, the orbits cut concavities into the anterolateral sides of the frontals.

Figure 10 Frontal of hatchling P. lujiatunensis (IVPP V15451).

(A) Left lateral view. (B) Right lateral view. (C) Ventral view. (D) Dorsal view. (E) Posterior view. (F) Anterior view. cc, cerebral cavity; fro, frontal ossicle; lsas, laterosphenoid articular surface; nas, nasal articular surface; om, orbital margin; paas, parietal articular surface; poas, postorbital articular surface. Scale bar represents 5 mm.

Juvenile (2-year old) braincase description

The juvenile specimen (IVPP V22647) proved difficult to segment due to poor preservation and obliterated sutures.

The skull has undergone uniaxial dorsoventral compression giving it a blocky appearance with an extremely flattened skull roof (Fig. 1C). The lack of a rostral also means that it appears rectangular in lateral view. Unlike the other specimens, the occipital surface of the juvenile is orientated posteriorly with no ventral inclination. It measures 76 mm (rostral—OC) by 30 mm (basipterygoid processes—frontal).

The foramen magnum is oval, with a flattened dorsal margin (Fig. 11D). Whether this is due to the taphonomic deformation is unclear. The FM is made up of 50% exoccipitals (laterally), approximately 35% supraoccipital (dorsally) and 15% basioccipital (ventrally).

Figure 11 Segmented braincase of a juvenile P. lujiatunensis (IVPP V22647).

(A) Lateral view. (B) Ventral view. (C) Dorsal view. (D) Posterior view. (E) Anterior view. bo, basioccipital; bpc, basisphenoid-parasphenoid complex; fr, frontal; ls, laterosphenoid; pa, parietal; pp, paroccipital processes; pr, prootic. Scale bar represents 10 mm.

The basioccipital contacts of the juvenile include the basisphenoid anteriorly and the paroccipital processes posterodorsally. As in the hatchling, the basioccipital forms the posterior half of the braincase floor. It measures 14 mm anteroposteriorly and 16.6 mm across its widest point. The basal tubera are poorly preserved compared to those of the hatchling and adult specimens. The tubera are approximately 5 mm long and are separated by what appears to be a very shallow medial groove (Fig. 12E: btg), but the depth of this feature may be the product of the taphonomic deformation and compression. Ventrally, the tubera have a loose, rounded L shape and no strong orientation (Fig. 12C: bt). Posteriorly, they are dorsoventrally compressed and ventrally extend no further than the OC (Figs. 11D, 12A, 12B and 12E).

Figure 12 Basioccipital of juvenile P. lujiatunensis (IVPP V22647).

(A) Left lateral view. (B) Right lateral view. (C) Ventral view. (D) Dorsal view. (E) Posterior view. (F) Anterior view. boc, basioccipital condyle; bsas, basisphenoid articular surface; bt, basal tubera; btg, basal tubera groove; cdn, condylar neck; eoas, exoccipital articular surface; ng, neural groove. Scale bar represents 10 mm.

The OC of the juvenile Psittacosaurus specimen is dorsoventrally flattened and, like the hatchling, is made up entirely of the basioccipital (Fig. 11D: bo). Whether this compression is due to post-mortem deformation or was present in vivo is not clear. As both the FM and OC are oval in the juvenile specimen and round in the other two specimens, the former is most probable. The OC measures 7.7 mm wide by 5.4 mm tall. There is a well pronounced condylar neck which exhibits ventral constriction. Similar to the OC of the hatchling, the neck restricts proximal to the condyle making it anterodorsally compressed and relatively thin in lateral view (Fig. 12: cdn).

A shallow groove on the dorsal surface of the basioccipital marks the ventral-most margin of the FM (Fig. 11D). Because the braincase is fused, the dorsal surface of the basioccipital is hard to distinguish. However, it is clear that the majority of the dorsal surface of the basioccipital is taken up by contact points for the two exoccipitals (Fig. 12D: eoas). The basioccipital seems to taper ventrolaterally at these surfaces, meaning it appears triangular when viewed posteriorly or anteriorly (Figs. 12E and 12F). The basioccipital constitutes 60% of the basal tubera, with the basisphenoid contributing approximately 20% more than it did in the hatchling.

The basisphenoid contacts the prootic dorsally and the basioccipital posteriorly. The pterygoid articular surface sits on the anterior-most surface of the basipterygoid processes (Fig. 13F: ptas). The BPC measures 30.8 mm anteroposteriorly by 20 mm across the widest point of the basipterygoid processes. As with the hatchling, the boundary between the basisphenoid and parasphenoid of the juvenile is completely obscured by extreme sutural fusion. The basipterygoid processes project anterolaterally from the main body of the basisphenoid and diverge at an angle of approximately 70°. They measure approximately 8.7 mm in length and expand distally. The juvenile parasphenoid is complete and the cultriform process exhibits minimal deformation (Fig. 13: cp). The cultriform process is long, thin and troughed. It measures 23 mm in length and, as in the hatchling, it contributes to approximately two-thirds of the total parabasisphenoid length. The paracultriform troughs are more angled and abrupt than the curving troughs of the hatchling (Fig. 13: pct). The sella turcica sits on the dorsal surface of the parabasisphenoid, posterior to the cultriform process (Fig. 13D: st). It is oval when viewed dorsally. The carotid foramina lie lateroventrally to the sella turcica (Figs. 13A and 13B: cfo). The basisphenoid contributes to a larger proportion of the basal tubera (approximately 45%) than in the hatchling specimen, although the location of the boundary remains the same.

Figure 13 Basisphenoid of juvenile P. lujiatunensis (IVPP V22647).

(A) Left lateral view. (B) Right lateral view. (C) Ventral view. (D) Dorsal view. (E) Posterior view. (F) Anterior view. boas, basioccipital articular surface; bpp, basipterygoid process; bt, basal tubera; cfo, carotid foramen; cp, cultriform process; pct, paracultriform trough; pras, prootic articular surface; ptas, pterygoid articular surface; st, sella turcica. Scale bar represents 10 mm.

The supraoccipital of the juvenile contacts the exoccipitals dorsolaterally and the parietal dorsally. It measures 9.8 mm at its widest point (ventral margin). The boundaries of the supraoccipital are inferred as they were unclear, making it hard to segment. It is anteroposteriorly triangular in shape (Fig. 14). The supraoccipital contributes to the dorsal portion (approximately 25%) of the FM (Fig. 11D: so). The supraoccipital lies at the midpoint between the adult’s thin, rod-like supraoccipital and the relatively large, plate-like supraoccipital that contributes to much of the posterior surface of the hatchling’s skull. The posterior semicircular canals are situated in the paroccipital processes rather than the supraoccipital as in the hatchling.

Figure 14 Supraoccipital of juvenile P. lujiatunensis (IVPP V22647).

(A) Posterior view. (B) Anterior view. eoas, exoccipital articular surface; fm, foramen magnum; paas, parietal articular surface. Scale bar represents 10 mm.

The paroccipital process contacts the basioccipital ventrally, the supraoccipital medially, the parietal dorsally and the prootic anteriorly. Most of the paroccipital processes are missing and fractured. Only the medial-most section of paroccipitals remains (Fig. 15: pop). A length to width ratio for the processes cannot be calculated as they are so incomplete. The exoccipitals make up approximately two thirds of the FM (Fig. 11D). Cranial nerve exits X–XII are located high up just below the lateral expansion of the exoccipitals which is similar to the hatchling. The location of the exit of the vena capitis dorsalis from the paroccipital processes is unclear due to poor preservation. A section of the posterior semicircular canal sits within the paroccipital.

Figure 15 Paroccipital processes of juvenile P. lujiatunensis (IVPP V22647).

(A) Left lateral view. (B) Right lateral view. (C) Posterior view. (D) Anterior view. boas, basioccipital articular surface; fm, foramen magnum; pop, paroccipital process; pras, prootic articular surface; sccp, semicircular canal pathway; soas, supraoccipital articular surface. Scale bar represents 10 mm.

The laterosphenoid of the juvenile contacts the prootic posteriorly, the parietal posterodorsally and the frontal anterodorsally. It measures 15.4 mm anteroposteriorly and has a width of 6.8 mm. It is morphologically very similar to the hatchling, differing only in being anteroposteriorly elongated. It is flared and triangular in cross section near the posterior boundary (Fig. 16). The laterosphenoid head is located anteriorly and is dorsoventrally flattened in comparison to the blocky central head of the hatchling (Fig. 16A: lsh). A process extends down from the laterosphenoid creating the anterodorsal corner of the exit for CN V.

Figure 16 Laterosphenoid of juvenile P. lujiatunensis (IVPP V22647).

(A) Right lateral view. (B) Right medial view. (C) Ventral view. (D) Dorsal view. (E) Posterior view. (F) Anterior view. fras, frontal articular surface; lsh, laterosphenoid head; paas, parietal articular surface; pras, prootic articular surface; CN V, dorsal margin of trigeminal nerve. Scale bar represents 10 mm.

The prootic contacts the paroccipital process posteriorly, the parietal posterodorsally, the laterosphenoid anteriorly, and the basisphenoid ventrally. The boundaries between the prootics, paroccipital processes and the parietal were difficult to infer. They measure 12.8 mm in height, assuming that the parietal-prootic contact is roughly correct. The prootics create the posterior and ventral walls of CN V with the preprootic strut making up the anterior margin (Fig. 17: pps). Posterior to the notch for CN V the prootic thickens and a series of anteroposterior ridges extend across the medial surface. The median ridge seen in the hatchling specimen does not appear to be so prominent in the juvenile. CN VII exits through the prootic posterior to this thickened strut (Figs. 17C and 17D). There is also a second opening next to CN VII which extends into the inner ear. On the lateral surface of the prootic there is a “V” shaped ridge extending anteroposteriorly.

Figure 17 Prootics of juvenile P. lujiatunensis (IVPP V22647).

(A) Left lateral view. (B) Right lateral view. (C) Left medial view. (D) Right medial view. (E) Ventral view. (F) Dorsal view. (G) Posterior view. (H) Anterior view. bsas, basisphenoid articular surface; eoas, exoccipital articular surface; lsas, laterosphenoid articular surface; paas, parietal articular surface; pps, preprootic strut; CN V, trigeminal nerve; CN VII, facial nerve. Scale bar represents 10 mm.

The parietal is contacted anteriorly by the frontals, anteroventrally by the laterosphenoids, lateroventrally by the prootics, posterolaterally by the squamosals and posteroventrally by the supraoccipital. It also contacts the parietal posteroventrally, but this cannot be seen because of damage and poor preservation. The contacts between the parietal, paroccipital processes and the prootics are inferred due to damage and deformation. It measures 16.3 mm anteroposteriorly and approximately 25 mm across the anterior-most margin. It is flattened, and a shallow sagittal crest has formed (Fig. 18D: sc). The apparent flatness is likely due to either taphonomic deformation or the early stage in development. Although the posterior portion of the parietal is broken and damaged, a small parietal shelf can be inferred (Fig. 18: ps?). The upper temporal fenestrae create concavities in the lateral margins of the parietal. The parietal borders the posterior margin of the upper temporal fenestra where they meet the squamosal distally.

Figure 18 Parietals of juvenile P. lujiatunensis (IVPP V22647).

(A) Left lateral view. (B) Right lateral view. (C) Ventral view. (D) Dorsal view. (E) Posterior view. (F) Anterior view. fras, frontal articular surface; lsas, laterosphenoid articular surface; pras, prootic articular surface; ps?, potential parietal shelf; sc, sagittal crest; soas, supraoccipital articular surface. Scale bar represents 10 mm.

The frontal contacts the parietal posteriorly, the nasal anteriorly, the laterosphenoid posteroventrally, the postorbital posterolaterally and the prefrontal anterolaterally, marked by a notch (Fig. 19D: no). The frontals measure 30.6 mm anterodorsally and 36.5 mm transversely across the widest point. Unlike those of the hatchling, the juvenile’s frontals are flattened, with only the anterior section of the cerebral concavities being present on the ventral surface (Fig. 18C: cc). The frontal of the juvenile is sub-triangular, flaring posteriorly, with the widest point being the contact with the postorbital (Fig. 19D: poas). The hourglass shape of the anterior region of the brain is preserved on the ventral surface of the frontal. The margin between the brain cavity and the orbits is well defined. The small frontal ossicles that were present on the hatchling specimen are no longer evident. A shallow frontal crest extends medially down the dorsal surface, where the two frontals meet in the sagittal suture. Laterally and parallel to this crest are two shallow ridges, although whether these are a product of taphonomic deformation is unclear. The orbits occupy shallow lateral concavities into the ventral surface of the frontal (Fig. 19: om).

Figure 19 Frontals of juvenile P. lujiatunensis (IVPP V22647).

(A) Left lateral view. (B) Right lateral view. (C) Ventral view. (D) Dorsal view. (E) Posterior view. (F) Anterior view. cc, cerebral cavity; fc, frontal crest; lsas, laterosphenoid articular surface; nas, nasal articular surface; no, notch; om, orbital margin; paas, parietal articular surface; poas, postorbital articular surface. Scale bar represents 10 mm.

Adult braincase description

The specimen IVPP V12617 was approximately 10 years old at the time of death (Zhao et al., 2013). It was previously referred to H. houi but that taxon is synonymized by most with P. lujiatunensis (Sereno, 2010; Hedrick & Dodson, 2013).

The cranium has a rounded anterodorsal surface, with a domed skull roof (Fig. 1E). It measures 146 mm in length (OC- rostral) by 60 mm in height. The skull has undergone some dorsoventral compression (Hedrick & Dodson, 2013), but the OC appears to have been comparatively unaffected by dorsoventral deformation and so we take the braincase to be negligibly deformed. The occipital surface of the skull is orientated posteroventrally, as is the case in the hatchling, but this orientation may be a result of the compression.

The circular foramen magnum is made up of the exoccipitals laterally (60%), supraoccipital dorsally (10%) and basioccipital ventrally (30%) (Fig. 20D).

Figure 20 Segmented braincase of adult P. lujiatunensis (IVPP V12617).

(A) Lateral view. (B) Ventral view. (C) Dorsal view. (D) Posterior view. (E) Anterior view. bo, basioccipital; bpc, basisphenoid-parasphenoid complex; fr, frontal; ls, laterosphenoid; os, orbitosphenoid; pa, parietal; pp, paroccipital processes; pr, prootic; so, supraoccipital. Scale bar represents 20 mm.

The basioccipital contacts the basisphenoid anteriorly and the exoccipitals laterodorsally. It is complete, with minimal to no deformation. The element measures 26.4 mm anteroposteriorly and 32 mm across its widest point. The basioccipital takes up approximately 30% of the margin of the FM, which is double that of the younger specimens. The posterior portion of the basioccipital forms the OC and the anterior section contributes to the basal tubera. The basioccipital contributes to approximately 50% of the basal tubera, with the basisphenoids making up the other half. The basal tubera are dorsoventrally and mediolaterally larger than the OC which differs from those of the hatchling which are approximately half the size of the condyle. The distinct groove separating the basal tubera is smaller and shallower relative to the main body of the basioccipital than those in the hatchling (Fig. 21E: btg). The basal tubera are anteroposteriorly flattened and are orientated mediolaterally (Fig. 21C: bt). The basioccipital is deeper than those of the younger individuals.

Figure 21 Basioccipital of adult P. lujiatunensis (IVPP V12617).

(A) Left lateral view. (B) Right lateral view. (C) Ventral view. (D) Dorsal view. (E) Posterior view. (F) Anterior view. boc, basioccipital condyle; bsas, basisphenoid articular surface; bt, basal tubera; btg, basal tubera groove; cdn, condylar neck; eoas, exoccipital articular surface; ng, neural groove. Scale bar represents 10 mm.

The OC is heart-shaped (Fig. 20D: boc) because the FM indents the dorsal margin. It is composed entirely of the basioccipital, as is the case with the younger specimens described here. The basioccipital-exoccipital contact was very well fused and so some sections were inferred. The basioccipital-exoccipital contact is higher up and closer to the flaring processes than in the younger specimens (Fig. 20D). This means that the basioccipital contributes to the base of the paroccipital processes, although cranial nerves X–XII still exit from the base of the exoccipitals. The OC measures approximately 14.3 mm wide by 11.9 mm deep. When viewed ventrally, the OC is triangular (Fig. 20B). Similar to the condylar neck of the hatchling and juvenile, there is some ventral constriction. This mature specimen differs from those in that the point of greatest constriction occurs closer to the basal tubera than to the OC.

The ventral margin of the FM is inscribed deeper into the dorsal surface of the basioccipital than in the younger specimens (Fig. 21D: ng) and widens anteriorly over the tubera. The dorsal surface of the basioccipital is triangular in outline. The contact separating the basioccipital and basisphenoid is visible as a transverse groove extending across the basal tubera (Figs. 20A and 20B).

The basisphenoid-parasphenoid complex of the adult contacts the prootic dorsally and the basioccipital posteriorly (Fig. 20). It measures approximately 24 mm anteroposteriorly and 35.9 mm across the basipterygoid processes. The basisphenoid is well preserved but, as with the other specimens, cannot be differentiated from the parasphenoid due to sutural fusion. It is a dorsoventrally deeper and more robust element than in the younger individuals. The basipterygoid processes measure approximately 14.3 mm in length and are ventrally projected compared to those of the younger individuals (Fig. 22: bpp). They diverge at an angle of approximately 44° making the paracultriform troughs thin and angular. The distal expansion of the basipterygoid processes is more extreme in IVPP V12617. The cultriform process is broken and incomplete, and only the tall, thin and blade-like portion, which is also present in the hatchling, is preserved (Fig. 22: cp). No prominent trough appears along the dorsal surface of the process as it does in the hatchling and juvenile specimens, but it may not be preserved due to damage. The sella turcica differs from those of the younger specimens in being more compact and sub-diamond in shape (Fig. 22D: st). This leads on to the cerebral carotid artery canal (Figs. 22A and 22B: cfo). These carotid foramina are positioned anteriorly compared to the younger specimens.

Figure 22 Basisphenoid of adult P. lujiatunensis (IVPP V12617).

(A) Left lateral view. (B) Right lateral view. (C) Ventral view. (D) Dorsal view. (E) Posterior view. (F) Anterior view. boas, basioccipital articular surface; bpp, basipterygoid process; bsr, basisphenoid recess; bt, basal tubera; cfo, carotid foramen; cp, cultriform process; pct, paracultriform trough; pras, prootic articular surface; ptas, pterygoid articular surface; st, sella turcica. Scale bar represents 10 mm.

The supraoccipital contacts the parietal anteriorly and the paroccipitals ventrolaterally; but, it should be noted that the fused and obscured sutures make the exact points of contact difficult to determine. It measures 12.3 mm dorsoventrally and, at 6.8 mm wide, is the thinnest supraoccipital in this study. It only contributes to approximately 10% of the FM (Fig. 20D: so), a much smaller proportion than in the younger skulls. The supraoccipital appears extremely reduced, forming a small rod-like nuchal crest dorsal to the FM (Fig. 23). The dorsal extension of the supraoccipital forms a ridge that extends dorsally from the FM to the ventral edge of the parietal. Unlike the supraoccipital of younger specimens, it appears that it does not sit in contact with the hindbrain. The semicircular canals do not invade the supraoccipital.

Figure 23 Supraoccipital of adult P. lujiatunensis (IVPP V12617).

(A) Left lateral view. (B) Right lateral view. (C) Ventral view. (D) Dorsal view. (E) Posterior view. (F) Anterior view. eoas, exoccipital articular surface; paas, parietal articular surface. Scale bar represents 10 mm.

The paroccipitals contact the basisphenoid ventrally, prootic anteriorly, parietal dorsally and supraoccipital mediodorsally. Distally, the paroccipital processes support the squamosals and contact the quadrate. These processes flare out from the FM in a pair of wing-like processes that expand laterally into large tabs (Fig. 20: pp). They measure 59.3 mm in length by 19.6 mm in height giving a ratio of nearly 1:3. The processes are anteroposteriorly compressed (Fig. 24). When viewed dorsally, the pair has a concave posterior surface (Figs. 24E and 24F) as the distal portions extend posterolaterally. Cranial nerves X–XII exit medioventrally through the exoccipitals and, unlike in the hatchling, are not visible posteriorly. Unfortunately, these foramina are hard to distinguish and, as a result, only two can be seen. These foramina are positioned more laterally, slightly lower, and further from the lateral expansion of the paroccipital processes than in the younger specimens. Semicircular canal foramina can be seen on the anterior surface (Fig. 24H). The exoccipitals make up approximately 60% of the FM margin.

Figure 24 Paroccipital processes of adult P. lujiatunensis (IVPP V12617).

(A) Left lateral view. (B) Right lateral view. (C) Left medial view. (D) Right medial view. (E) Ventral view. (F) Dorsal view. (G) Posterior view. (H) Anterior view. boas, basioccipital articular surface; fm, foramen magnum; pop, paroccipital process; pras, prootic articular surface; CN X–XII, approximate location of the foramen transmitting the vagus nerve, accessory nerve, and the hypoglossal nerve. Scale bar represents 10 mm.

The laterosphenoid contacts the prootic posteroventrally, the frontal anterodorsally and the orbitosphenoid anteromedially. The laterosphenoids of the adult are robust and nearly triangular in cross section (Fig. 25). They measure 25.9 mm anteroposteriorly and 14.5 mm wide anteriorly. Like those of the juvenile, the laterosphenoid is posteroanteriorly elongated, and the head is dorsoventrally flattened and located on the anterior surface (Figs. 25A and 25B: lsh). The laterosphenoid is concave medially and convex laterally. In lateral and medial views, it appears to taper anteriorly (Figs. 25A and 25B), but when viewed anteriorly it appears to taper posteriorly (Figs. 25E and 25F). The laterosphenoid only contributes to the dorsal-most margin of the anterolaterally oriented foramen for CN V.

Figure 25 Laterosphenoid of adult P. lujiatunensis (IVPP V12617).

(A) Left lateral view. (B) Right lateral view. (C) Ventral view. (D) Dorsal view. (E) Posterior view. (F) Anterior view. fras, frontal articular surface; lsh, laterosphenoid head; osas, orbitosphenoid articular surface; pras, prootic articular surface. Scale bar represents 10 mm.

The thin posterior margin of the orbitosphenoid forms the articular surface for the laterosphenoid (Fig. 26: lsas). The orbitosphenoid measures approximately 27 mm dorsoventrally. This left orbitosphenoid is the only example preserved in our specimens. It is mediolaterally thin and delicate in this mature specimen making it understandable why this element is rarely preserved in smaller individuals. The optic nerve exits medially where it inscribes a concavity in the orbitosphenoid (Fig. 26).

Figure 26 Orbitosphenoid of adult P. lujiatunensis (IVPP V12617).

(A) Left lateral view. (B) Left medial view. (C) Posterior view. (D) Anterior view. lsas, laterosphenoid articular surface; CN II, optic nerve. Scale bar represents 10 mm.

The boundaries of the prootics are difficult to distinguish, particularly the dorsal-most contacts. Inferred contacts include the basisphenoid ventrally, laterosphenoid anterodorsally, parietal posterodorsally and the paroccipital processes posteriorly. It is possible that it does not contact the parietal and an anterior portion of the exoccipitals inserts between the two elements. The prootic of the adult is a tall and relatively robust element (Fig. 27) that measures 29.2 mm in height and 13.5 mm in width. The bone surrounding the semicircular canals is damaged, and so boundaries between the prootic and exoccipital are obliterated. The anterior canal and parts of the horizontal semicircular canal are well entombed in the prootic, which differs from the hatchling (Fig. 27G: sccp). The anterior sections of the semicircular canals can be seen exiting the posterior surface of the prootic (Fig. 27G). The medial surface of the prootic lacks both the medial ridge of the hatchling and the transverse ridges of the juvenile. With the exception of the cranial nerves, the medial surface is concave and smooth. CN V is almost entirely enclosed in the prootic due to the presence of a large, robust preprootic strut (Fig. 27: pps).

Figure 27 Prootics of adult P. lujiatunensis (IVPP V12617).

(A) Left lateral view. (B) Right lateral view. (C) Left medial view. (D) Right medial view. (E) Ventral view. (F) Dorsal view. (G) Posterior view. (H) Anterior view. bsas, basioccipital articular surface, eoas, exoccipital articular surface; lsas, laterosphenoid articular surface; pps, preprootic strut; sccp, semicircular canal pathway; CN V, trigeminal nerve. Scale bar represents 10 mm.

The parietal contacts include the frontal anteriorly, prootic anteroventrally, paroccipitals posteroventrally and the supraoccipital posteriorly. The parietal lies posteriorly on the dorsal surface of the braincase and contributes approximately 50% of the braincase roof (Fig. 20C: pa). The posterior position of the parietal differs from the parietal of the hatchling specimen which sits dorsally to the brain. Although the posterior section of the parietal is missing, the preserved parietal suggests that a small shelf projected over the occipital surface. A midline ridge spans the height of the posterior surface of the parietal shelf (Fig. 28E: pmr) forming the supraoccipital articular surface. A sagittal crest is a prominent feature of the adult parietals (Figs. 28A, 28B and 28D: sc) and spans the length of the dorsal surface. The mid-lateral portions of the parietal taper laterally to form the margins of the upper temporal fenestra.

Figure 28 Parietals of adult P. lujiatunensis (IVPP V12617).

(A) Left lateral view. (B) Right lateral view. (C) Ventral view. (D) Dorsal view. (E) Posterior view. (F) Anterior view. eoas, exoccipital articular surface; fras, frontal articular surface; lsas, laterosphenoid articular surface; pmr, parietal midline ridge; sc, sagittal crest; soas, supraoccipital articular surface; sqas, squamosal articular surface. Scale bar represents 10 mm.

Frontal contacts include the parietal posteriorly, laterosphenoids ventrally and the nasals anteriorly. It is unclear where the prefrontal contact would sit (Fig. 29D: no?). The frontal measures 54 mm anteroposteriorly and 51.5 mm across its widest point and is dorsoventrally very thin (<5 mm) along the midline. Much like the frontal of the juvenile, this element is relatively flat in the adult (Fig. 29). It appears sub-rectangular in dorsal view. Ventrally, the hourglass impression of the anterior region of the brain is present and only the anterior-most section of the cerebrum impressions is preserved. These impressions are shallower than the bulbous impressions of the hatchling specimen. A low, blunt sagittal crest is present on the dorsal surface of the frontal (Fig. 29: fc). As is the case with the juvenile frontal, the orbit wall is orientated in the coronal plane obscuring the orbit from lateral view. When viewed dorsally, the orbital margin (Fig. 29: om) inscribes anterolateral concavities into the sides of the frontal.

Figure 29 Frontals of adult P. lujiatunensis (IVPP V12617).

(A) Left lateral view. (B) Right lateral view. (C) Ventral view. (D) Dorsal view. (E) Posterior view. (F) Anterior view. cc, cerebral cavity; fc, frontal crest; lsas, laterosphenoid articular surface; nas, nasal articular surface; no, notch; om, orbital margin; paas, parietal articular surface; poas, postorbital articular surface. Scale bar represents 10 mm.

Discussion

Skull proportions

A comparison of the three skulls (Table 1) reveals that several elements remain relatively isometric throughout growth. The salient allometric changes that accompany growth include approximately 40% elongation of the paroccipital processes (relative to basal skull length), a dramatic reduction in supraoccipital height (approximately 50% smaller relative to basal skull length) and width (approximately 90% smaller relative to basal skull length), approximately 50% reduction in width of the laterosphenoids and the frontals (relative to basal skull length), and over 50% increase in the height of the basal tubera (relative to basal skull length). Measurements taken from the juvenile braincase do not correspond closely to those taken from the other individuals in that the height to width ratios of the elements do not follow the same trend as in the other two specimens. This is most likely due to taphonomic deformation (mainly dorsoventral compression) or unavoidable inaccuracies during segmentation because of suture obscurity or obliteration.

Table 1 Element sizes in relation to base skull length (%) and the total % change between hatchling and adult.

	Hatchling	Juvenile	Adult		
	Element (mm)	Skull (mm)	%	Element (mm)	Skull (mm)	%	Element (mm)	Skull (mm)	%	Total % change	
BO w	5.8	23.6	24.58	16.6	86.5	19.19	32	143.7	22.27	−2.31	
BO l	5.4	23.6	22.88	14	86.5	16.18	26.4	143.7	18.37	−4.51	
bt h	1.2	23.6	5.08	4.4	86.5	5.09	16.4	143.7	11.41	6.33	
bt w	2.9	23.6	12.29	8.8	86.5	10.17	16	143.7	11.13	−1.15	
OC w	3	23.6	12.71	6.8	86.5	7.86	14.3	143.7	9.95	−2.76	
OC h	2	23.6	8.47	5	86.5	5.78	11.9	143.7	8.28	−0.19	
BSp w	7.3	23.6	30.93	20	86.5	23.12	35.9	143.7	24.98	−5.95	
BSp l	5.9	23.6	25.00	15.9	86.5	18.38	27.8	143.7	19.35	−5.65	
cp l	6.9	23.6	29.24	23.5	86.5	27.17	11.1	143.7	7.72	−21.51	
POP l	6.2	23.6	26.27	0	86.5	0.00	59.3	143.7	41.27	15.00	
POP h	2.8	23.6	11.86	0	86.5	0.00	19.6	143.7	13.64	1.78	
SOC h	3.6	23.6	15.25	0	86.5	0.00	12.3	143.7	8.56	−6.69	
SOC w	8.7	23.6	36.86	9.8	86.5	11.33	6.8	143.7	4.73	−32.13	
Lsp w	4.2	23.6	17.80	6.8	86.5	7.86	14.5	143.7	10.09	−7.71	
Lsp l	5.2	23.6	22.03	15.4	86.5	17.80	25.9	143.7	18.02	−4.01	
Pr l	7.1	23.6	30.08	12.8	86.5	14.80	29.2	143.7	20.32	−9.76	
Pr w	3.9	23.6	16.53	0	86.5	0.00	13.5	143.7	9.39	−7.13	
Par l	7.4	23.6	31.36	16.3	86.5	18.84	39.8	143.7	27.70	−3.66	
Fr l	11.2	23.6	47.46	30.6	86.5	35.38	54	143.7	37.58	−9.88	
Fr w	15.8	23.6	66.95	36.5	86.5	42.20	51.5	143.7	35.84	−31.11	
Note:

bo, basioccipital; bsp, basisphenoid; bt, basal tubera; cp, cultriform process; fr, frontal; lsp, laterosphenoid; OC, occipital condyle; par, parietal; pop, paroccipital process; pr, prootic; soc, supraoccipital. w, width; l, length; h, height.

The orientation and morphology of the basioccipital component of the basal tubera changes during growth. The basal tubera of the hatchling are oriented in the sagittal plane (Figs. 30A and 30B). The tubera of the juvenile are starting to develop into the anteroposteriorly compressed tubera of the adult basioccipital, but they still retain some of the mediolateral compression observed in the basioccipital of the hatchling, which creates a loose “L” shape (Figs. 30C and 30D). The robust basal tubera of the adult are oriented transversely (Figs. 30E and 30F). As the individual grows, we also see a larger contribution of the basisphenoid to the basal tubera, which might explain this morphological change. The basioccipital of adult individuals of Bagaceratops rozhdestvenskyi differs from that of P. lujiatunensis in that it contributes very little to the basal tubera (Maryańska & Osmólska, 1975). Unlike the basal tubera of Yinlong (Han et al., 2016) and pachycephalosaurids (Maryańska, Chapman & Weishampel, 2004), the basisphenoid contribution of P. lujiatunensis is not visible in caudal view.

Figure 30 Morphological changes of the basioccipital contribution to the basal tubera in ventral view and the location of the basal tubera within the braincase.

(A) Schematic view and (B) location of basal tubera of hatchling. (C) Schematic view and (D) location of basal tubera of juvenile. (E) Schematic view and (F) location of basal tubera of adult. Not to scale.

During growth, we see an increase in the size of the basal tubera relative to the rest of the braincase. The height to width ratio of the tubera also changes from 1:2 in the two younger specimens, to 1:1 in the adult. The tubera of the adult are wide, plate-like, but clearly bilobate. Neoceratopsians such as Liaoceratops, Auroraceratops and Archaeoceratops have a singular clear plate beneath the OC with a shallower median cleft (Dodson, You & Tanoue, 2010). Basal tubera act as ligament and muscle attachment sites, stabilizing the head on the neck. This enlargement of the attachment area may be attributed to the expansion of the parietal shelf and subsequent increase in relative weight of the skull. The growth of the basal tubera likely permitted strengthening of the lateroflexion muscles, such as the m. longissimus (Ostrom, 1961), required for movement of the neck in conjunction with the developing skull. This expansion is observed in other non-avian dinosaurs (Carpenter, 1982; Jacobs et al., 1994; Huebner & Rauhut, 2010).

The OCs studied here, with the possible exception of the juvenile (likely due to sutural obscurity resulting in segmentation imprecision), are made up entirely of the basioccipital, with no evident contribution from the exoccipitals. Several descriptions of Psittacosaurus suggest this is the case (Sereno et al., 1988; Sereno, 1992; Xu, 1997; You & Xu, 2005; Zhou et al., 2006; Brinkman et al., 2001; You, Tanoue & Dodson, 2008). This contradicts Averianov et al. (2006) and Sereno (2010), both of whom suggested that the exoccipitals likely contributed to the OCs, as is the case in some other dinosaurs including more derived ceratopsians.

The length ratio of the basioccipital and basisphenoid (disregarding the cultriform process) is approximately 1:1 and this remains relatively static throughout growth (Table 1). Further, the cultriform process stays the same length relative to total basisphenoid length. The cultriform process also moves dorsally as the basisphenoid deepens throughout ontogeny. The anterolateral projection of the basipterygoid processes observed here are present in all psittacosaurids (Sereno et al., 1988; You, Tanoue & Dodson, 2008; Dodson, You & Tanoue, 2010) and similar rostrally projecting processes are present in several basal neoceratopsians including Archaeoceratops (Dodson, You & Tanoue, 2010) and Protoceratops (Brown & Schlaikjer, 1940; Chinnery & Weishampel, 1998; Dodson, You & Tanoue, 2010). Conversely, posteroventrally projecting basipterygoid processes are found in the Leptoceratopsidae (Chinnery & Weishampel, 1998; Ott, 2007) and the Ceratopsidae (Hatcher, Osborn & Marsh, 1907; Dodson & Currie, 1990; Chinnery & Weishampel, 1998). Throughout growth, these basipterygoid processes become progressively more ventrally projecting. The angle at which the basipterygoid processes diverge also decreases from 81° to 44°. This may be to accommodate the dorsoventral expansion of the skull during growth. When viewed dorsally, the sella turcica changes from triangular to sub-diamond-like in shape.

The supraoccipital contributes to the dorsal margin of the FM in each specimen reported here. This is also the case in Protoceratops (Brown & Schlaikjer, 1940), Leptoceratops (Sternberg, 1951) and Bagaceratops (Maryańska & Osmólska, 1975). The supraoccipital bisecting the exoccipitals and contributing to the dorsal margin of the FM is also thought to be a juvenile character within Ceratopsidae (Gilmore, 1917; Lehman, 1989), having been recorded in immature specimens of Triceratops (Goodwin et al., 2006; Horner & Goodwin, 2006), Brachyceratops (Gilmore, 1917) and Chasmosaurus (Lehman, 1989). This contribution to the FM is then lost in adult ceratopsids as the exoccipitals exclude the supraoccipital from the roof of the braincase (Hatcher, Osborn & Marsh, 1907; Brown & Schlaikjer, 1940; Dodson & Currie, 1990; Forster, 1996; Dodson, Forster & Sampson, 2004). The supraoccipital in P. lujiatunensis appears to undergo a reduction in size and dramatic change in shape. The large, thin, plate-like supraoccipital of the hatchling reduces in size (relative to basal skull length) with age and becomes rod-like with a wider ventral base. The midline crest running down the posterior face of the supraoccipital in the hatchling and adult skulls most likely separates an attachment site for epaxial muscles, as is the case for other ceratopsians such as Triceratops (Goodwin et al., 2006). Unlike the other individuals, the juvenile supraoccipital shows no evidence of a caudal midline ridge, but this area is incomplete. While the supraoccipital forms most of the occipital surface of the hatchling skull, much of this surface of the adult skull appears to be taken up by the parietal. As P. lujiatunensis grows, the contribution of the supraoccipital to the FM decreases. A more extreme version of this, where the supraoccipital is fully excluded from the FM by the exoccipitals, is documented during growth of ceratopsids such as Triceratops (Goodwin et al., 2006). The posterior semicircular canal of P. lujiatunensis is situated in the supraoccipital of the hatchling, whereas it remains within the confines of the paroccipital processes in the older specimens. It is important to note that compression of the adult skull (Hedrick & Dodson, 2013) may have affected sutural boundaries and segmentation accuracy in this area. While You & Xu (2005) and Taylor et al. (2017) also report a heavily reduced supraoccipital in IVPP V12617, other reconstructions of adult Psittacosaurus have depicted the supraoccipital as being wider than it is tall (Sereno, 1987; Sereno et al., 1988; Zhou et al., 2006). It is therefore possible that the supraoccipital of the adult was more laterally extensive than is apparent in this specimen.

The paroccipital processes expand laterally during growth of P. lujiatunensis. The small, rectangular processes of the hatchling become long and slightly distally flared. These gracile, long and dorsoventrally narrow elements are similar to those of basal neoceratopsians such as Montanoceratops (Chinnery & Weishampel, 1998), Leptoceratops (Sternberg, 1951), Bagaceratops (Maryańska & Osmólska, 1975) and Proceratops (Brown & Schlaikjer, 1940). Ceratopsid exoccipitals tend to be stouter, more robust, and highly flared distally, contacting the squamosal to support the large parieto-squamosal frill (Brown & Schlaikjer, 1940; Ostrom & Wellnhofer, 1990; Forster, 1996; Chinnery & Weishampel, 1998). Goodwin et al. (2006) note that the paroccipital processes of the juvenile Triceratops appear to anticipate the structural requirements of adulthood and are already flared, in contact with the ventral surface of the squamosal, and laterally expanded to form the buttress for the frill. The expansion of the paroccipital processes of P. lujiatunensis increases the surface area of the attachment site of the m. obliquus capitis magnus, for lateral and dorsoventral movement of the head. This might have been necessary to accommodate the weight of the mature skull with developed parieto-squamosal shelf and flared jugals, a hypothesis also used to explain the large exoccipitals in ceratopsids (Goussard, 2006). This muscular growth may also be linked to the posited postural shift during ontogeny (Zhao et al., 2013) and the associated possibility of a change in feeding mechanism—that is, grazing juveniles develop into facultative browsers. Much like those of Bagaceratops (Maryańska & Osmólska, 1975), the contribution of the exoccipitals to the OC of P. lujiatunensis is minimal/ absent. The exoccipitals do not contact each other, being separated ventrally by the basioccipital and dorsally by the supraoccipital, as in other basal ceratopsians such as Yinlong (Han et al., 2016). In basal Neoceratopsia, the exoccipitals also contribute to the ventral portion of the FM, excluding the basioccipital (You & Dodson, 2004).

The laterosphenoid undergoes some morphological change and becomes more robust as the individual grows. Laterally, it is triangular and transforms to become anteroposteriorly elongated. The laterosphenoid head displaces anteriorly as the laterosphenoid elongates. The m. pseudotemporalis superficialis, m. tensor periorbitae and m. levator pterygoideus are assumed to have had attachment sites on the laterosphenoid (Holliday, 2009).

The semicircular canals are loosely held within the prootic of the hatchling, which is partially why it was difficult to segment. The prootic develops into a robust element and the semicircular canals of the two older specimens are encased firmly within. The foramen for cranial nerve V becomes orientated more anteriorly through growth along with the development of a pronounced preprootic strut. A small foramen in the middle of the hatchling and juvenile prootics is thought to be for transmitting cranial nerve VII. If this is the case, this is shared with Montanoceratops (Chinnery & Weishampel, 1998) and Bagaceratops (Maryańska & Osmólska, 1975).

Initially, the parietal and frontal are convex in accordance with the round skull shape which is also observed in a juvenile specimen of Bagaceratops (Maryańska & Osmólska, 1975). These bones become flatter and, as observed in Coombs (1982), a sagittal crest develops with age, a clear indication of strengthening adductor mandibulae jaw muscles (Sereno, Xijin & Lin, 2010). This crest is also present in several basal ceratopsians including Yinlong (Han et al., 2015), Liaoceratops (Xu et al., 2002), Archaeoceratops (You & Dodson, 2003) and Bagaceratops (Maryańska & Osmólska, 1975). Neither the hatchling described here nor the juvenile specimen of Bagaceratops described by Maryańska & Osmólska (1975) have a developed parietal crest, so a defined sagittal crest is likely an adult character. During growth, a small parietal shelf extends posteriorly over the occipital surface. This ontogenetic “frill” growth is also seen in basal neoceratopsians such as Protoceratops (Maryańska & Osmólska, 1975; Fastovsky et al., 2011) and Bagaceratops (Maryańska & Osmólska, 1975). Ceratopsids, such as Triceratops, develop a large frill during growth, but an incipient frill is already present in juvenile specimens (Goodwin et al., 2006; Currie et al., 2016). As the parietal expands during growth, the supraoccipital shrinks (relative to basal skull length) and the contact between the parietal and the paroccipital processes extends across the full width of the processes. The fronto-parietal sutural contact of Protoceratops migrates posteriorly with growth (Maryańska & Osmólska, 1975), which differs from P. lujiatunensis whose contact appears to remain relatively stationary.

The frontals contact the same elements throughout growth. Substantial negative allometry can be observed in the width of the frontals during growth (Table 1). As in juvenile Bagaceratops (Maryańska & Osmólska, 1975), they remain dorsally convex and contribute to the overall roundness of the skull roof. Unlike Bagaceratops (Maryańska & Osmólska, 1975), the frontal of P. lujiatunensis contributes to a large portion of the orbital margin. A frontal crest develops with age. The juvenile has well defined, deep bulbous concavities where the cerebral hemispheres sat. The cerebral depressions on the frontal become shallower and less defined during growth, likely due to increasing distance between the brain and the braincase through maturation (Jerison, 1973). In cross section, the small convexities present on the midline of the dorsal surface of the frontals appear to be separate, individual ossicles. There are several indicators that these are not an artifact of the scanning process. Firstly, they mirror each other either side of the midline ridge, displaying almost perfect bilateral symmetry. Had the specimen not been set perfectly in the scanner, this would be off. Secondly, the cross section through these ossicles shows the same texture and contrast values as the rest of the skull. We are also confident that these ossicles are not broken sections from other elements due to the perfect circular shape and lack of any broken or damaged surfaces. The midline symmetry also supports this. We considered the possibility that these structures had a function similar to that of an egg-tooth and helped the baby Psittacosaurus to hatch. This is, however, unlikely as the egg-tooth of archosaurs (including putative examples in other non-avialan dinosaurs) is generally found on the tip of the snout (Rahn, Ar & Paganelli, 1979; García, 2007). It is unlikely but possible that they could represent a pathology or a mutation as they have not been observed in other juvenile ceratopsians, including hatchling Psittacosaurus (Coombs, 1982). They could be epiossifications much like those seen on the parieto-squamosal frill, but these frontal ossicles are lost during growth.

Semicircular canals

The semicircular canal pathways are not clear in the hatchling IVPP V15451 and, consequently, some sections have been inferred (Fig. 31A). This could be due to cartilaginous surroundings at a young age, and this is more likely than simply poor preservation, as none of the reviewed hatchling specimens has a complete housing for the inner ear. A theorized pathway has been created using what remains of the hatchling’s bony labyrinth (Fig. 31A) and the morphology of the canals in the two older specimens.

Figure 31 Left semicircular canals displaying the angle between the ASC and the PSC.

Hatchling in (A) lateral and (B) dorsal views. Juvenile in (C) lateral and (D) dorsal views. Adult in (E) lateral and (F) dorsal views. ASC, anterior semicircular canal; PSC, posterior semicircular canal; HSC, horizontal semicircular canal. Inferred sections in red. Not to scale.

The semicircular canals of the juvenile are dorsoventrally short. It is possible that this was the case in vivo, but it is quite likely a result of the taphonomic compression that has affected the skull as a whole. Because this stunted morphology spans the entire inner ear, the height relationship between the anterior and posterior semicircular canals is preserved. The posterior semicircular canal is approximately two-thirds the height of the anterior canal (Fig. 31C).

The semicircular canals of the adult are well preserved and maintain their in vivo morphology. Much like those of the juvenile, the anterior semicircular canals are at least twice as large as the posterior semicircular canals (Fig. 31E).

Lateral (horizontal) semicircular canal

It has been determined that Psittacosaurus changed its posture during growth. Juveniles (including IVPP V16902) are reconstructed as quadrupeds, based on the similar lengths of their forelimbs and hindlimbs (Zhao et al., 2013). There is a lot of evidence for bipedality in adult Psittacosaurus. The forelimbs are half the length of the hindlimbs (Osborn, 1923; Maryańska & Osmólska, 1975; Sereno, 1990; Zhao et al., 2013) and observed modifications of the pelvis suggest that they were able to support more weight cranially (Chinnery, 2004). Similarly, Psittacosaurus display bipedal character states for all osteological correlates set out by Maidment & Barrett (2014). Bipedality in adult Psittacosaurus is also supported by the robustness and mobility of the forelimbs, which led Chinnery (2004) to suggest that they could be used to manipulate the surrounding environment. Further evidence comes from Maidment, Henderson & Barrett (2014) who calculated the center of mass for adult Psittacosaurus was located dorsal to the hindfoot, allowing for bipedal locomotion. Psittacosaurus are frequently reconstructed, when adult, as facultative bipeds, dropping their bodies and arms to the ground when feeding, for example, but rearing high to detect danger or feed on leaves high on a tree. This postural shift was confirmed by bone histological analysis of forelimb and hindlimb bones, in which the femur of juveniles and adults shows evidence of faster growth than the humerus (Zhao et al., 2013). Bone histology also demonstrated that the shift in posture likely occurred during the third year of life, emphasizing the negative allometric growth of the forelimb relative to overall body length as the animal became more and more adapted to bipedality during ontogeny, after age 4 (Zhao et al., 2013).

We seek to test this idea by reference to the lateral semicircular canal in the three growth stages, from hatchling to adult. The linkage between the lateral semicircular canal orientation and head posture has been discussed for nearly a century. Although some studies advocated that the orientation of the lateral (horizontal) semicircular canal (LSC) is variable at both intraspecific and interspecific scales (Duijm, 1951; Taylor, Wedel & Naish, 2009; Marugán-Lobón, Chiappe & Farke, 2013; Coutier et al., 2017), the LSC is generally considered to remain earth-horizontal when the head is in its “natural” alert position (Lebedkin, 1924; Witmer et al., 2003, 2008; Chatterjee & Templin, 2004; Sereno et al., 2007; Araujo et al., 2017; Coutier et al., 2017; Benoit et al., 2017; Schellhorn, 2018).

When working with semicircular canals, it is important to be sure that the method of measuring orientation is plausible, and indeed our data may contribute to the debate. When Witmer et al. (2003) used the orientation of the LSC to assume the resting posture of the pterosaur head, critics (Taylor, Wedel & Naish, 2009) pointed out that in living tetrapods the LSC is not always held horizontal. For example, these authors noted that extant animals often hold their heads oriented so the LSC is tilted upwards by 12° in monkeys, 16° in rabbits, 20° in guinea-pigs and domestic cats, and 22° in humans (Graf, De Waele & Vidal, 1995, Spoor & Zonneveld, 1998). In birds, Duijm (1951) showed that in most species the LSC was held horizontal, but in some species, values were as much as 20° below or 30° above horizontal. Similarly, Marugán-Lobón, Chiappe & Farke (2013) showed that LSC orientation was not a reliable indicator of head orientation at rest. However, in studies of modern xenarthrans (Coutier et al., 2017) and rhinos (Schellhorn, 2018), the LSC was near enough horizontal to provide a good indicator of head orientation.

We propose to use our study of Psittacosaurus as a test of these opposing views. The angles between the plane of the LSC and the palatal plane were calculated in our study, following the method in Schellhorn (2018). Our analysis shows that the angles are very variable in the ontogenetic series of P. lujiatunensis, but they show a clear trend of decline during ontogenetic growth. The angle is 38° in the hatchling stage, and changes to 25° in the juvenile, and reaches 15° in the adult. Based on the isometric growth line of P. lujiatunensis (Zhao et al., 2014), we can estimate the hatchling specimen is less than 1 year old, and the juvenile specimen is approximately 2 years old. The adult specimen is 10 years old, according to bone histology. Under the assumption that the plane of the LSC is earth-horizontal, we reconstruct the head posture of Psittacosaurus in different growth stages (Fig. 32). In our reconstruction, the nose-down head posture is quite obvious in the hatchling stage, which implies quadrupedal locomotion as the angle between the skull and neck vertebrae would be too high for a bipedal stance (Coutier et al., 2017). The nose rises up in the juvenile stage, and points forward in the adult stage. These changes of head posture are consistent with the previous study on posture shift from quadrupedal to bipedal during growth in Psittacosaurus (Zhao et al., 2013). The larger angle present in the hatchling specimen would not be congruent with a bipedal stance (Coutier et al., 2017).

Figure 32 Head posture if LSC is parallel to the ground.

(A) Hatchling. (B) Juvenile. (C) Adult. Not to scale.

Cranial nerves X–XII

There have been many interpretations and configuration of the foramina exiting through the exoccipitals below the paroccipital processes in early ceratopsians (Averianov et al., 2006; Brown & Schlaikjer, 1940; Chinnery & Weishampel, 1998; Forster, 1996; Maryańska & Osmólska, 1975; Xu, 1997; Brinkman et al., 2001). Here, CNXII3 is the posterior-most foramen and is the largest in the hatchling and juvenile specimens. It is doubtful that CN X and CN XI would have individual exits, as they are confluent in ceratopsids and so it seems unlikely that they would separate just to converge again (Brown & Schlaikjer, 1940; Averianov et al., 2006). As noted by Averianov et al. (2006), the foramen for CN XII1+2 is small and can be overlooked, which is why often Psittacosaurus can sometimes be mistakenly described as having only two foramina in the exoccipital (Xu, 1997). This is most likely the case for the adult described here, as only two cranial nerve foramina can be seen exiting the exoccipital, unlike the hatchling and juvenile specimens, which have three clear, separate foramina (Fig. 33). Note that poor contrast values in the adult scan made segmentation difficult in this region.

Figure 33 Schematic drawing of foramina for cranial nerves X–XII.

(A) Hatchling. (B) Juvenile. (C) Adult. Not to scale.

Conclusion

Braincases are often neglected in cranial descriptions and ontogenetic studies because they are believed to exhibit little variation and are often inaccessible. While some braincase elements grew in an isometric fashion, many grew at different rates (Table 1). Salient ontogenetic changes in the braincase of P. lujiatunensis (Fig. 34) include:The basal tubera expand dorsoventrally and laterally.

The angle of divergence of the basipterygoid processes drops from 81° to 44°.

The supraoccipital appears to undergo a dramatic reduction in size from a large, plate-like element to a transversely compressed rod. One caveat is that the accuracy of segmentation in this area may have been affected by damage.

The small sub-rectangular paroccipital processes of the hatchling laterally expand and become long and “strap-like.”

The laterosphenoid becomes anteroposteriorly elongated and the laterosphenoid head displaces rostrally.

A sagittal crest develops along the midline of the parietal.

A small parietal shelf develops.

The width of the frontals relative to basal skull length decreases dramatically during growth.

The angle of the lateral semicircular canal decreases from 38° to 15°. This is the first evidence to illustrate the variation of orientation of the LSC in the ontogenetic growth of a species of dinosaur, and it confirms that, at least for this genus, the LSC orientation is well aligned with expectations of head posture. The head was held nose-down in quadrupedal hatchlings and more horizontal in bipedal adults. This shift is in accordance with matching of the OC and the vertebral column.

Figure 34 Prominent ontogenetic changes in the braincase of Psittacosaurus lujiatunensis.

(A) Lateral view. (B) Posterior view. (C) Hatchling and (D) adult braincases in situ. Not to scale.

Braincases are generally more conservative than other areas of the skeleton that are associated with highly adaptive functions, and so can be valuable in a phylogenetic context (Bakker, Williams & Currie, 1988; Coria & Currie, 2002). Element characteristics that change dramatically during growth should be excluded from a phylogenetic analysis as ontogenetic variation may be misinterpreted as phylogenetic variation. Elements that remain similar in relative size or morphology throughout growth may prove to be useful phylogenetic characters and should be investigated further. In this case, the proportions of the basioccipital, paroccipital height, parietal length and basal tubera width should be explored in a phylogenetic context, as the relative growth of these elements appears to be close to constant (Table 1).

This is the first time that frontal ossicles have been recorded in basal ceratopsians. The function and origins of these ossicles are unknown. More examples and further analysis of these structures are required to make any reliable assumptions.

Supplemental Information

Supplemental Information 1 Element sizes in relation to base skull length (%).

bo, basioccipital; bsp, basisphenoid; bt, basal tubera; cp, cultriform process; fr, frontal; lsp, laterosphenoid; OC, occipital condyle; par, parietal; pop, paroccipital process; pr, prootic; soc, supraoccipital. W, width; l, length; h, height.

Click here for additional data file.

Firstly, we thank Yun Feng, the technician at the IVPP, for scanning the specimens. We thank the editor and reviewers for their manuscript suggestions and improvements. We thank James Brown for his artistic assistance with the Psittacosaurus silhouettes. Finally, we would like to thank Tom Davies, the Palaeobiology Laboratories Manager, whose assistance with Avizo and the 3D models was invaluable.

Institutional abbreviations

IVPP Institute of Vertebrate Paleontology and Paleoanthropology, Chinese Academy of Sciences, Beijing, China.

Additional Information and Declarations

Competing Interests

Author Contributions

Data Availability

The authors declare that they have no competing interests.

Claire M. Bullar conceived and designed the experiments, performed the experiments, analyzed the data, contributed reagents/materials/analysis tools, prepared figures and/or tables, authored or reviewed drafts of the paper, approved the final draft.

Qi Zhao conceived and designed the experiments, performed the experiments, analyzed the data, contributed reagents/materials/analysis tools, authored or reviewed drafts of the paper, approved the final draft.

Michael J. Benton authored or reviewed drafts of the paper, approved the final draft.

Michael J. Ryan conceived and designed the experiments, authored or reviewed drafts of the paper, approved the final draft.

The following information was supplied regarding data availability:

The specimens are held within the collections of the Institute of Vertebrate Paleontology and Paleoanthropology in Beijing, China.

Hatchling id: IVPP V15451

Juvenile id: IVPP V22647

Adult id: IVPP V12617

The original scan data is available at any time by application to the Collection Manager, Fang Zheng ( zhengfang@ivpp.ac.cn), Curator contact: curator@ivpp.ac.cn, or Collection office email: bbg@ivpp.ac.cn at IVPP, Beijing.

Claire Bullar (2019): Ontogenetic sequence of Psittacosaurus lujiatunensis braincases. https://doi.org/10.5523/bris.3l2y3f6tdptcr2rcseeei6b26h.

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
