# Peer review of "Ontogenetic braincase development in Psittacosaurus lujiatunensis (Dinosauria: Ceratopsia) using micro-computed tomography"

_PeerJ, doi:10.7717/peerj.7217_

## Round 0.1 · original submission · Major Revisions

Dear Claire,

I have now received three reviews of your manuscript submitted to PeerJ. The reviewers recommend a number of improvements that should be addressed before re-submission, including a closer attention to the literature (e.g., Lautenschlager & Hübner, 2013).
As per PeerJ policies (https://peerj.com/about/policies-and-procedures/#data-materials-sharing), all the raw data have to be made available in a permanent public repository prior to formal acceptance.
Please, together with your unmarked revised manuscript, provide a marked-up copy as well as a document explaining how you have addressed each of the points raised by the reviewers.

Best regards,
Fabien

·

Basic reporting

The manuscript is well written, clear and comprehensive, but not unnecessarily long or wordy. The anatomical description of the three specimens is very detailed and complete. Descriptions are accompanied by very professionally looking figures, which show the anatomy comprehensively and help with the understanding of the otherwise often complex anatomy of the braincase elements.
The introduction provides a good overview of the topic and previous research. Methods and specimen details are sufficiently outlined. The discussion puts the findings for the most part in a good context with previous research. However, a comparison with ontogenetic series of other dinosaur species is missing. I do realise that the manuscript focusses on ceratopsian dinosaurs, for which little data exists, however, there are several examples of reported ontogenetic stages in other ornithischian dinosaurs (e.g. Evans et al. 2009; Lautenschlager & Hübner 2013). A more detailed discussion of these in the context of the current study would be very appropriate.



Evans, D. C., Ridgely, R., & Witmer, L. M. (2009). Endocranial anatomy of lambeosaurine hadrosaurids (Dinosauria: Ornithischia): a sensorineural perspective on cranial crest function. The Anatomical Record, 292(9), 1315-1337.

Lautenschlager, S., & Hübner, T. (2013). Ontogenetic trajectories in the ornithischian endocranium. Journal of Evolutionary Biology, 26(9), 2044-2050.

Experimental design

The experimental design is appropriate and all methodological approaches have been defined/explained in sufficient detail. The research question is stated in the introduction and clearly defined/addressed.

Validity of the findings

The anatomical descriptions are detailed and comprehensive. Findings are supported by appropriate figures and data tables. Conclusions are provided and linked to the outlined research question.

Additional comments

This is a very well-executed and detailed study of three ontogenetic stages of the basal ceratopsian Psittacosaurus. The manuscript is well written, appropriately detailed and accompanied by professional and clear figures. My biggest gripe with the manuscript is the lack of discussion of other ontogenetic series of (ornithischian) dinosaur brain cases/endocasts. I would like to see some comparisons with published studies and how Psittacosaurus differs from other ornithischians. This would allow making inferences beyond a single species and possibly provide some more general patterns for brain/braincase development in dinosaurs more broadly.
A second point, where this study could be improved would be to present a summary figure showing the findings in a morphological context. I wonder if it would be possible to schematically present the ontogenetic changes similar to Figure 3a in Lautenschlager & Hübner 2013 for the braincase (I am aware, that I am shamelessly suggesting one of my own publications, but the authors might find this useful),
In addition, I have a few minor, more specific comments and suggestions listed below. These, however, should be relatively easy to incorporate.
All in all, this is a nice contribution to our knowledge of ornithischian development and ontogenetic patterns, and I am sure it will be of interest to a wider audience of palaeontologists interested in these topics. Based on the well-written and professionally illustrated manuscript, I will recommend publication pending minor revisions.


List of additional comments:
Line 40-42: It would be good to add a brief (e.g. 1-2 sentences) explanation on the relevance of chaoyangsaurids for readers without intimate knowledge on early ceratopsian taxonomy.

Lines 44-45: The two paragraphs feel quite unrelated and it would be helpful to link them together thematically (e.g. psittacosaurs important for understanding of early ceratopsian evolution in particular with regards to locomotion/posture…)

Lines 64-65: As above, these two paragraphs are topically unconnected. Possibly mention the importance of both psittacosaurs and ontogeny to address questions of heterochrony in ceratopsian evolution or similar?

Lines 74-79: Move this to the methods.

Lines 85: I do agree that the models are “visually striking”. Nevertheless, remove this qualifier as it is not a methodological step and sounds somewhat unprofessional.

Line 113: Abbreviations for basioccipital (BO) missing. Compare with other elements for consistency.

Line 135: You refer to the basisphenoid-parasphenoid complex here, but it is referred to as the basisphenoid only in the corresponding figures.

Lines 161-162: The reference should be to figure 4x not figure 5.

Lines 182-183: In figure 6 only the hypoglossal nerve is indicated to pass through the foramen.

Line 209: “Main trunk” is ambiguous. Do you mean the cochlear duct?

Line 229 “…which is the location of the cerebral hemispheres” – Better “… which represent the space occupied by the cerebral hemispheres of the forebrain”. On that note, are there any impressions of vascular structures or similar recognisable which would hint at a close contact of the brain and the frontals. Either way, this should be mentioned.

Line 264: “of the basioccipital”

Line 306: Remove “unfortunately”. I share the feeling but try to keep a neutral language.

Line 322: “Difficult” is probably more appropriate than “hard”. Is this due to the fusion or the resolution of the scan?

Line 404: “sub-diamond in shape” is unclear. “Circular” or “ovoid” might be a better description?

Line 423: “Nerves X-XII exit medio-ventrally through the exoccipitals…” – showing the foramen for the nerves in the figure would be helpful.

Line 470: Remove “at”.

Line 506-507: Would it be expected that the contribution of the individual bones to the occipital condyle changes? If the condyle is entirely formed by the basioccipital in the hatchling this would mean a relative reduction of the element if the with growth. I know that the contribution of different elements is variable in different species, but is there an example for changes within an ontogenetic sequence?

Line 543: Is this shrinking relative to other elements or in absolute size? This also applies to several other occasions in the text, where a reduction in size is mentioned. I assume this is a relative reduction compared to the rest of the skull, but make sure this is clear.

Line 547-548: “Such large, well-defined hemispheres are unusual within ornithischians.”- I would disagree with this statement. Evans et al. (2009) and Lautenschlager & Huebner (2013) show a few examples with well defined cerebral hemispheres.

Evans, D. C., Ridgely, R., & Witmer, L. M. (2009). Endocranial anatomy of lambeosaurine hadrosaurids (Dinosauria: Ornithischia): a sensorineural perspective on cranial crest function. The Anatomical Record, 292(9), 1315-1337.
Lautenschlager, S., & Hübner, T. (2013). Ontogenetic trajectories in the ornithischian endocranium. Journal of Evolutionary Biology, 26(9), 2044-2050.

Lines 573-575: “They are frequently reconstructed, when adult, as facultative bipeds, dropping their bodies and arms to the ground when feeding, for example, but rearing high to seek danger or feed on leaves high on a tree.” – This sentence is rather convoluted and would benefit from rephrasing. Also, not convinced the adults are seeking danger…

Lines 592-604: Excellent discussion/summary of the current debate regarding head orientation using the LSC. I would stress a bit more that recent studies, such as Schellhorn 2018 actually find a correspondence between hypothesised posture and observations of living specimens.


Figure 4: Abbreviation “pct” not explained in the caption.

Figure 25: Abbreviation “pps” not in the figure.

Figure 30: The outlines are a bit abstract. It would help if a small picture of the specimen or the digital model with the location of the basal tubera is added.

Figure 31: I would like to see a dorsal view of the inner ears as well so that the outline of the LSC and the angle between the ASC and the PSC becomes clear.

Figure 32. This will require some inferences and artistic skills, but would it be possible to add little silhouettes of the whole animal below each skull showing the full body posture?

Table 1: It would be useful to add another table or additional columns to the existing one showing the percentage increase between the three specimens

·

Basic reporting

The manuscript is well written, but with some errors in structure, spelling, citation and reference.

Experimental design

no comment

Validity of the findings

This is a good manuscript that firstly report the ontogenetic information of the psittacosaur braincase.

Additional comments

Dear Editor, Dear Authors,
This is a good manuscript describing the first ontogenetic information of the psittacosaur braincase. The manuscript is well written, but with some errors in spelling, citation, and so on. I list them as below. Besides these, I would like to highlight some other issues that should be addressed before this manuscript goes to press:

1) As mentioned by the authors, the Lujiatun psittacosuars are deformed more or less, as the juvenile V12617 and the adult V22647; and the hatchling IVPP V15451 is possibly deformed too. Normally, the skull of psittacosaurs is subrectangular in lateral views without a distinct posterior inclination; so that the adult V22647 is more deformed than the juvenile V12617. It is not correct to consider the posteriorly-inclined cranial structure as the natural condition by the authors, as in lines 360-361 “The occipital surface of the skull is orientated postero-ventrally, as in the case in the hatchling”.
And also, these deformed skulls would impact on the orientation of semicircular canals. The authors should consider how to exclude this influence, especially when the orientation be used to index the head position. The authors should include more undeformed fossils in this study, at least scan one more adult skull to check this influence in V22647. As we know, psittacosaurs are common in many Chinese collections, should also in IVPP.

2)Line 58, About the isotope dating of Lujiatun deposits, Jiang et al. (2011) did not work on the radiometric age. And also, the 124 Ma is replaced by an updated age of 126.0 Ma by Chang et al. (2017). The reference is here, Chang S-C, Gao K-Q, Zhou C-F, Jourdan F. 2017. New chronostratigraphic constraints on the Yixian Formation with implications for the Jehol Biota. Palaeogeography, Palaeoclimatology, Palaeoecology, 487:399-406.

3)Lines 74-79, this paragraph is only about three skulls, should be included in “Materials & Methods”.
4)Line 95, ZMNH is Zhejiang Museum of Natural History, Hangzhou, China. Please correct it.
5)Lines 97-99, the citation should be present in Reference Part.
6)Line 101, “Zhou, 2006” should be “Zhou et al., 2006”
7)Lines 96-102, “Systematic Palaeontology”, the authors should write out the whole thing, such as, paratype, referred specimens, and diagnosis. Alternatively, this part can be deleted.
8)Line114, “5.3 mm” is inconsistent with “5.4 mm” in table 1.
9)Line 125, “constructed” may be “constricted”
10)Lines 223 and 224, “the nasals” and “postorbitals” should be singular form.
11)Line 263, “The occiput” should be “The occipital condyle”; please delete “Psittacosaurus”, its taxonomy is already known in the Materals and methods Part. Do not need to repeat this again.
12)Line 305, “the prootic” should be plural form.
13)Line 312, “paroccipitals” should be singular form.
14)Line 320, “the paroccipital processes” should be singular form.
15)Line 322, “the parietals” is singular form.
16)Line 330, “the frontal” and “the laterosphenoid” should be plural form.
17)Line 331, “the squamosal” should be plural form.
18)Line 334, “prootic” should be “prootics”
19)Lines 341, 342, “the nasals”, “the postorbitals” and “the prefrontals” should be singular form.
20)Line 535, “Holiday, 2009” is not present in the Reference part.
21)Line 553, “Anterior and posterior semicircular canals” should be deleted. This paragraph is about the general information of semicircular canals, not just for the two canals.
22)Line 563, “the anterior semicircular canal is approximately three times taller than the posterior one”, I did not find this feature in Fig. 31B. In Fig. 31B, the anterior canal is taller than the posterior one, but not too much as mentioned by the authors. Please check it again.
23)Line 570, “including” should be added in front of “IVPP V16902”
24)Lines 581-582, “(LSC, or ‘horizontal’ semicircular canal)” should be moved to Line 569, behind “Lateral semicircular canal”; “our” should be replaced by “the”
25)Lines 582-584, the sentence should be deleted, due to it is meaningless.
26)Lines 596-597, a repeat between “120 in humans” and ”220 in humans”.
27)Lines 597-598, the citations “Graf et al., 1995; Spoor & Zonneveld, 1998” are not in the Reference Part.
28)Line 601, “2016” should be corrected as “2017”
29)Line 621, “Averianov, 2006” should be replaced by “Averianov et al., 2006”
30)Line 627, the description about the number of nerve foramina in three skulls, is inconsistent with that shown in Fig. 33. Please check it.
31)Lines 652-663, this paragraph should be included in the Discussion Part.

With best regards,

Chang-Fu Zhou

Reviewer 3 ·

Basic reporting

Overview:
The article describes in detail the anatomy of the braincase from an ontogenetic series consisting of three individuals of Psittacosaurus lujiatunensis, a basal Ceratopsian dinosaur from the Barremian of Liaoyang Province, PRC. The anatomy of each element is described based on segmented CT data, and the anatomy of the reconstructed vestibular apparatus (semicircular canals) is described. Semicircular canal orientation is used to infer posture for differing ontogenetic stages of Psittacosaurus.

I would like to begin my review with a reminder that any comments here are intended to be constructive and I apologize in advance if any of my comments strike the readers as anything otherwise.

Basic Reporting

Figures are of good quality, and it is encouraging to see the large number of figures representing the elements in multiple views. It would assist the reader if there were a way to view 3D representations of the renderings. Perhaps 3D pdfs of the individual elements and whole cranial reconstructions could be provided?
In some places in the text, labelling additional features in some figure views would make it easier for the reader to follow the description. Suggestions for additional labels in the figures are made below in the detailed list of comments.
The description notes that segmentation of several of the elements was difficult and that sutural boundaries were inferred (i.e. in the juvenile braincase: the supraoccipital [line 296], basioccipital [line 271], prootic [line 321-322], parietal [line 333-334], in the adult braincase: the basioccipital-exoccipital contact [line 378], the supraoccipital [line 407] and others). I appreciated how the reconstructed areas of the vestibular apparatus were highlighted in red in Fig. 31. The same should be done for all of the contact surfaces that are inferred and have higher uncertainty than the other surfaces of the bones presented in the figures. This way the reader knows exactly whether a surface was observed or inferred.
The citations are somewhat sparse throughout the paper. Though not voluminous, a literature exists on the braincases of Ceratopsian dinosaurs, going back to Hatcher et al., (1907), which is not properly cited. A more recent publication dealing with the external anatomy of the basicranium of Psittacosaurus specifically, and its relationship to the basicranium of more derived ceratopsians is Dodson et al. (2010).
The discussion of ontogenetic changes in Psittacosaurus seems oddly divorced from the literature on ontogenetic change in other Ceratopsian dinosaurs. Extensive growth series exist for Protoceratops (Brown and Schlaikjer, 1940; Dong and Currie, 1993; Fastovsky et al., 2011), Bagaceratops (Maryanska and Osmolska, 1975), and, further afield, Triceratops (Horner and Goodwin, 2006). Less extensive growth series exist for Liaoceratops (Xu et al., 2002), Montanoceratops (Chinnery and Weishampel, 1998; Makovicky, 2010), and Chasmosaurus (Currie et al., 2016). Comparisons between known changes in the elements in these taxa and the patterns seen in this paper would be valuable. Although few of the previous descriptions look at endocranial elements, per se, at least the elements of the occiput and dorsal skull roof could be compared.
Also strangely omitted in the background or introduction is any reference to the previous work on juvenile Psittacosaurus, including Coombs (1982) an extensive description of perinatal Psittacosaurus cranial anatomy and ontogeny in a growth series of Psittacosaurus. Further comments on juvenile Psittacosaurus can be found in Sereno (2010). The introduction need not be vastly expanded, however some comments relative to the previous work on the ontogeny of ceratopsians and previous work on Psittacosaurus would help put the present addition into context.
The literature on the bipedal to quadrupedal transition in ornithischians in general and ceratopsians specifically is also undercited in this manuscript. Work on ceratopsian posture by Chinnery (2004) is directly relevant to the history of inferences of posture in Psittacosaurus and work relating histological features to posture in Chinnery and Horner (2007) may be useful (I leave the use of this second reference up to the authors).
The raw data was not included in the article. As the raw data represents CT data which some repositories view as a risk for unauthorized duplication (e.g. 3D printing), that is understandable. However, the authors should note the repository for the CT data and what institution qualified researchers should contact for access. Presumably the data are housed at IVPP. If not, the data must be placed in an appropriate repository where qualified researchers may access it in perpetuity. Again, it is acceptable for this repository to be curated, as in natural history museums where curators may limit access to specimens while they are under study, but the data should be accessible at some later date. Contacting the authors for data access is unacceptable and not in keeping with modern best-practices.
The structure is well organized. Each element described in a consistent pattern.
The authors should re-read the paper for small errors. I noted that in the institution list ZMNH was listed as “Zigong Museum of Natural History, Zigong, China”. This is incorrect as ZMNH was listed in the paper as the prefix for the holotype of Psittacosaurus lujiatunensis which is housed at the Zhejiang Museum of Natural History in Hangzhou, China. A few minor comments on organization, apparent omissions or mistakes in the text, and grammatical corrections are included in the list of line-by-line comments below.

Experimental design

Experimental Design
This descriptive paper follows the general pattern seen in many vertebrate paleontology articles. Methods for scanning were well described.
Graphing the relative size data of the three skulls as % basal skull length vs. basal skull length may improve the clarity of the comparison of dimensions between the three skulls and the discussion of isometric vs. allometric growth of different bones in the braincase.

Validity of the findings

Validity of the Findings
The conclusion needs to be better organized and slightly expanded in a few places. A common issue with papers that include much anatomical description is so much work goes into the description that the discussion and conclusion sections become neglected. This is something that I’ve done myself, and had reviewers guide me into improving my conclusions.
The discussion of changes in skull proportions would benefit from further comparisons to the ontogenetic series of other ceratopsians. How is Psittacosaurus different? Are there any commonalities. Line 486 mentions that the measurements from the juvenile braincase do not correspond to those from the other individuals. I am unsure what is meant by correspond in this context. If the intent is to say that the juvenile has proportions more unlike either the perinatal individual or the adult, then the sentence should be rephrased. I think that is the most likely intent, but one could also interpret that sentence to mean that the measurements in the juvenile are somehow using different landmarks, or different start and end points and so cannot be compared to the other two. If that is so, it needs further explanation.
The discussion of the lateral semicircular canal needs to be slightly expanded. A discussion of changes in posture through ontogeny for Psittacosaurus requires citation of the relevant literature on the posture of Psittacosaurus, including recent work by Maidment, Barrett and colleagues on ornithischian posture, with citations of the myological modelling, osteological correlate, and center of mass lines of inquiry that they pursued. Citations of older morphometric work is also needed (Chinnery, 2004; Chinnery and Horner, 2007). Citations of reconstructions of Psittacosaurus as a facultative biped mentioned in lines 574-575 are also needed. Even if these are popular works, or outreach sections of museum websites, they are worthy of documentation as they are the product of the thinking of experts in many cases. Also, in this discussion, the expected orientation of the head at neutral position for a quadruped or biped is not mentioned. It is unclear what the meaning of the different alert postures are. There are more specific comments following.
The conclusion section is very difficult to follow. It needs to be reorganized, with clear connections between statements of data from the paper and conclusions. One could describe the logical structure as “we found A, therefore we can say B”. I have more specific comments below, but the section needs to be largely restructured.
There were issues with several references, where Hailu You was referred to by his given name (Hailu) rather than his family name (You), as is traditional in citations. I have noted the corrections.
There are two findings that are quite strange and seem to be only briefly mentioned.
1) The discussion of the supraoccipital in the adult Psittacosaurus seems to give the element an anatomy that is unlike any other previous reconstruction of that bone in adult Psittacosaurus in the literature. Given that the authors state that the borders of the supraoccipital are unclear in the adult specimen (line 407) would not a morphology incorporating flat material lateral to the thin caudally directed lamina which is currently assigned to the supraoccipital by the authors, be the more conservative approach? A supraoccipital bearing a ridge but being composed of bone on either side of that ridge is consistent with our understanding of the supraoccipital from other ceratopsians where it is known (e.g. Dodson et al., 2010; Morschhauser, 2012), and from non-ceratopsian neornithischians (e.g. Galton, 1974), which are our closest comparisons from an earlier diverging lineage as all basal ceratopsians lack well-preserved supraoccipitals. All previous interpretations of Psittacosaurus have proposed a larger supraoccipital than is reconstructed here (Dodson et al., 2010; Sereno, 1987; 2010; You et al., 2008).
2) The additional ossifications reported in the hatchling associated with the frontal (line 234-237) are described and characterized in two sentences as neomorphic bones similar to the epiossifications seen in derived ceratopsids, from which the structures reported here are separated by no small phylogenetic distance, as well as over 30 million years of time. It seems unlikely at first glance that these would have anything to do with epiquamosals and epiparietals, but the authors have not really argued that case. A more thorough discussion of these structures is necessary. The assertion of the authors would be strengthened by a discussion of evidence to disprove alternative hypotheses for these structures (for example, are they artifacts of the scanning process? Are they broken fragments of bone from elsewhere in the skeleton, or elements from another taxon that were taphonomically displaced to their current location? Could this specimen have been a perinate and associated with eggshell? How do these structures look under a light microscope?).
Overall the changes I am requesting are largely for clarity, expanding the review of past work slightly, and organization/grammatical in nature.

Additional comments

Line-by-line comments
Line 4 University of Bristol does not currently list Qi Zhao on its website (which seems quite good as reporting research affiliates and the like). The relationship with Bristol should be clarified (Visiting faculty perhaps?)
Line 19: Abstract states that ontogenetic sequences are relatively rare among ceratopsian dinosaurs. Rare compared to what? Dinosaur ontogenetic series are generally rare, but within dinosaurian, Ceratopsia is one of the better represented clades, with six or seven general with extensive or partial ontogenetic sequences.
Line 24 age of the Yixian Formation should be mentioned here, either in years, or to the age (Barremian)
Line 39: Authors use the informal term Psittacosaur, which is permissible, but it always struck me as odd. This entire group the psittacosaurs are all members of a single genus, why not just call it Psittacosaurus and be done with it?
Line 39-40 Citation needed for the statement “a diverse and geographically widespread suborder of ornithischian dinosaurs.” Perhaps use Dodson et al., 2004 and the relevant biogeography chapters from the Dinosauria 2nd edition to support.
Line 41 Citation mentioning Sereno, 2000; Morschhauser, 2012 should also mention Han et al., 2017 which recovers Psittacosaurus as basal most in some MPTs.
Line 51 P. sibiricus does not appear to be much smaller than the largest P. lujiatunensis.
Line 95 Institution incorrect. ZMNH M8137 is housed in the Zhejiang Museum of Natural History, Hangzhou, China
Line 118 It would improve reader understanding if the basal tubera grove (btg) were labelled on Fig 3C as well.
Line 180 Replace “tab-like” with a more descriptive term. A “tab” is not a shape (though many authors like to use it as such). If rectangular or square is the outline intended, then use those terms.
Line 183-184 “They [Nerve pathways X-XII] are visible when the braincase is viewed posteriorly.” It would be helpful to label these on Fig. 6 where the exoccipital is in posterior view.
Line 189 Change “vcd” to “vcd?” to conform with Fig. 6D.
Line 197-198 Label the portion of the laterosphenoid that makes up the dorsal boundary of Cranial Nerve V on relevant views in Fig. 7.
Line 210-211 For the sentence ending “…the prootic when viewed dorsally.” Add a citation of the portion of Figure 8 where the prootic is in dorsal view and cite the abbreviation for the pathway of the semicircular canal used in the figure.
Line 220-221 Label the indentations formed in the lateral margins of the parietal in Figure 9.
Line 235-237 “We are assuming they [frontal ossicles] are epiossifications much like those seen on the parietosquamosal frill...” Why? Seems like a big assumption. See my previous comment on how this section or a section discussing these features in the discussion needs to go test (even if only by thought experiment) alternative hypotheses for these structures before stating conclusions. Also, this sentence is more interpretation than description, and probably belongs in the discussion section anyway.
Line 260 The L-shape of the basioccipital tubera is unclear from the scans as presented in Fig. 12. Perhaps the direction or angle of the lighting of the scans in the virtual environment can be changed to highlight the rostral extension of the tubera better.
Line 261-262 “Posteriorly, they are almost entirely obscured by the occipital condyle.” In Fig. 12E the tubera seem to clearly sit lateral to the occipital condyle. Clarify this statement.
Line 269 “(Fig. 12: cdn)” This implies that there is a label on Fig. 12, but labels appear to be missing on Fig. 12B, 12C, 12D and 12E.
Line 273 Again, citing an abbreviation on Fig. 12D, but no labels are present on that figure section.
Line 283 Sentence beginning “The basisphenoid-paraspehenoid complex…” repeats something stated in line 280. Delete this sentence here
Line 295 Figure 14A is inverted relative to Fig. 14B. Since PeerJ technically has no pages, “white space” around figures becomes an aesthetic concern, not a space concern. The clarity gained from having both views in the same orientation more than makes up for the slightly less efficient use of space in the image.
Line 311 Label the posterior semicircular canal on the paroccipital figure.
Line 319 Label the antero-dorsal corner of the exit for CN V on Fig. 16.
Line 338 Label the small parietal shelf on Fig. 18
Line 352 It is unclear to me whether the frontal crest or the shallow parasagittal ridges are a product of taphonomic deformation.
Line 370 “The basal tubera are larger than the OC…” In what way are they larger? Dorsoventral depth? Etc.
Line 393 Reference Fig. 20 at the end of the sentence “…and the basioccipital posteriorly.”
Line 395 “It is a deeper…” Deeper in what way? Dorsoventrally? Mediolaterally? Specify in the sentence.
Line 399 Label the paracultriform ridge in Fig. 22.
Line 414-416 “Unlike the supraoccipitals of younger specimens it appears that it does not contribute to the surfaces of the cranial vault surrounding the brain.” This seems odd. The supraoccipital is involved in the external portion of the braincase wall in Protoceratops and Triceratops.
Line 422 “concave anterior surface” The anterior surface of the pop is convex on the lateral half of the process. Is this statement referring to the medial portion of the exoccipitals/opisthotics?
Lines 432-427 Label foramina for cranial nerves X-XII on Fig. 24. Also, go through the document and any place referring to these nerves make sure they are referred to as “cranial nerves X-XII” not “nerves X-XII”.
Line 436 “dorsal-most margin of the anteriolaterally orientated CN V” Change “orientated” to “oriented” Also this is a foramen margin, not a nerve margin. Also, label this border in Figure 25.
Line 456 Label the preprootic strut on more views in Fig. 27
Line 468 “It is unclear where the prefontal contact would sit.” My first inclination would be to place the transition from nasal contact to prefrontal contact at that break in slope in the sutural margin.
Line 475 Add reference to “Fig. 29: fc” at the end of the sentence “A low, blunt sagittal crest…”
Line 486 Explain what is meant by “correspond closely” here.
Line 506 Sereno (2010) also suggests that the exoccipitals contribute to the occipital condyle in juvenile Psittacosaurus. Cite somewhere here and adjust the sentences to include.
Line 511 It is important to note that the exoccipitals contribute significantly to the occipital condyle of derived Ceratopsian dinosaurs (most members of Neoceratopsia), so Averianov et al (2006) were on fairly solid ground when making that suggestion.
Line 520-522 The discussion of the supraoccipital has far more certainty that is warranted by the data. The adult specimen does not give clear indications of where the borders of the supraoccipital are and, given the fact that this adult specimen is known to be dorsoventrally compressed and sheared (Hedrick and Dodson, 2013), it is not the strongest ground to suggest dramatic shape changes from. These changes are possible at best, not even likely given the fact that both juvenile and adult skulls shown here are taphonomically distorted. Rephrase section with less certainty.
Line 529 How is the muscular growth linked to the postural change? Is it more difficult to hold a head essentially horizontally when bipedal vs when quadrupedal?
Line 547-548 Large well-define hemispheres first posited in Coombs, 1982. Do we have other perinatal ornithischians to compare with the current sample to see what shape the brains are at this stage?
Line 549-560 “likely due to thickening of the meninges and expansion of the brain through maturation.” Need to cite a reference indicating that the meninges thicken through ontogeny.
Line 569-570 “It is well-known that…” Even established features of non-ceratopsid ceratopsians are not well-known in the vert paleo community outside a small circle of ceratopsian workers, let alone proposed ontogenetic postural shifts. It has been proposed and supported by a few papers.
Line 572 Citation needed for sentence ending in “…lengths of their forelimbs and hindlimbs.”
Line 573 Citation needed for “Adults were always… half the length of the hindlimbs.”
Line 577 Citation needed for sentence beginning “Bone histology…adapted to bipedality during ontogeny, after age 4.”
Line 582 The sentence “The postural and bone histological analysis… and a definitely bipedal adult” is out of place. Move to later in the discussion.
Line 592 “It is important to be sure that the method is plausible, and indeed our data may contribute to resolving the debate.” This seems like an optimistic appraisal of the value of the data in question. While it certainly meaningfully contributes to the debate, it will only go so far in resolving it. Rephrase the sentence.
Line 611-614. What do these reconstructed habitual alter postures mean for the stance of Psittacosaurus? Is one more likely for a biped or quadruped? Why? Need to explicitly state what head postures one would expect for a quadruped or a biped and then state what the observed pattern means for Psittacosaurus gait. At the moment the reader does not see why the changes of head posture are highly consistent with the previous study on posture and locomotion. Also, please provide a single sentence summary of those results here. People shouldn’t have to look up another paper to understand what the big picture conclusions are when those conclusions are the basis of the argument. To be sure, I am not suggesting a detailed summary of Zhao et al., 2013, but the points most salient to the argument need to be made in this paper.
Line 627 The authors should explore the implications of the changes in the cranial nerve foramina in the exoccipital more. I feel like this section is too short. Phylogenetically, members of Ceratopsidae have only two openings, while protoceratopsids, basal neoceratopsians, and Psittacosaurus should have three.
Line 630-631 “The CT study reported here… post-processing of the scan data” This is a throw-away line. Many studies have shown this. Either cite a few, or just delete the sentence entirely and adjust the following sentence to be the first sentence in the paragraph.
Line 630-636 This entire paragraph is difficult to follow. I am unclear as to what the message of this paragraph is.
Line 641 “…the LSC orientation is well aligned with expectations of head posture” I am repeating a point, but the authors need to explicitly state what those expectations of head posture for different gaits are.
Line 643-644 “Braincases are often neglected…because they are believed to be static and unvarying” Citation needed. I always thought it was because they were either inaccessible inside articulated skulls or because they weren’t preserved in many disarticulated ones.
Line 645 “…many grew at different rates” Cite the data table which demonstrates this quite clearly.
Line 645-646 “Elements that remain similar in relative size or morphology throughout growth may prove useful phylogenetic characters” Based on what? Presumed consistency in morphology through ontogeny associated with consistent relative size? This sentence appears to be making a point, but it is incomplete at the moment. Complete the thought for what makes a good phylogenetic character and why braincases should be good character sources. The following sentence is also unclear. Clarifying it might help make this point.
Line 649-650 See above long comment on the frontal ossicles.
Line 651 “Such well defined hemispheres are unusual in ornithischians.” Citation needed.
Line 654 Change “(Hailu & Dodson, 2004)” to “(You & Dodson, 2004)” Also change reference. Author’s name is Hailu You (when presented in the western format of given name first, surname second).
Line 655 “This frill becomes large and ornate in ceratopsids.” Citation needed. Perhaps Dodson et al., 2004
Line 661-663 “Ceratopsid braincases appear very different as a whole, as do the individual elements, but the general configuration remains the same as in basal ceratopsians.” This sentence seems vague & unhelpful. Are there specific differences or similarities to highlight?
Just glancing over the references, I’m noticing a lack of DOIs for electronic resources. Those can expedite the process of locating electronic resources and I’d recommend including them in every electronically accessed reference unless PeerJ specifically forbids it. The authors should double-check their references to make sure there are no typos and that the reference format is consistent.
References cited
Brown, B. and E.M. Schlaikjer. 1940. The structure and function of Protoceratops. Annals of the New York Academy of Sciences 40:133–266.
Chinnery, B.J. 2004. Morphometric analysis of evolutionary trends in the ceratopsian postcranial skeleton. Journal of Vertebrate Paleontology 24:591-609.
Chinnery, B.J. and D.B. Weishampel. 1998. Montanoceratops cerorhynchus (Dinosauria: Ceratopsia) and relationships among basal neoceatopsians. Journal of Vertebrate Paleontology 18:569–585.
Chinnery, B.J., and J.R. Horner. 2007. A new neoceratopsian dinosaur linking North American and Asian taxa. Journal of Vertebrate Paleontology 27:625–641.
Coombs, W.P., Jr. 1982. Juvenile specimens of the ornithischian dinosaur Psittacosaurus. Palaeontology 25:89-107.
Currie, P.J., R.B. Holmes, M.J. Ryan, and C. Coy. 2016. A juvenile chasmosaurine ceratopsid (Dinosauria, Ornithischia) from the Dinosaur Park Formation, Alberta, Canada. Journal of Vertebrate Paleontology 36:2 DOI: 10.1080/02724634.2015.1048348.
Dodson, P., C.A. Forster, and S.D. Sampson. 2004. Ceratopsidae; pp. 494–513 in D.B. Weishampel, P. Dodson and H. Osmólska (eds.), The Dinosauria, second edition. California University Press, Berkeley California
Dodson, P., H. You, and K. Tanoue. 2010. Comments on the basicranium and palate of basal ceratopsians. Pp. 221-233 in M.J. Ryan, B.J. Chinnery-Allgeier, and D.A. Eberth (eds.). New Perspectives on Horned Dinosaurs: the Royal Tyrrell Ceratopsian Symposium. Indiana University Press, 624p.
Dong, Z. M., & Currie, P. J. (1993). Protoceratopsian embryos from Inner Mongolia, People's Republic of China. Canadian Journal of Earth Sciences, 30:2248-2254.
Fastovsky, D. E., Weishampel, D. B., Watabe, M., Barsbold, R., Tsogtbaatar, K. H., & Narmandakh, P. (2011). A nest of Protoceratops andrewsi (Dinosauria, Ornithischia). Journal of Paleontology, 85: 1035-1041.
Galton, P.M. 1974. The ornithischian dinosaur Hypsilophodon from the Wealden of the Isle of Wight. Bulletin of the British Museum of Natural History 25:1–152.
Hatcher J.B., O.C. Marsh, and R.S. Lull. 1907. The Ceratopsia. U.S. Geological Survey Monograph 49:1–300.
Hedrick, B. P., & Dodson, P. (2013). Lujiatun psittacosaurids: understanding individual and taphonomic variation using 3D geometric morphometrics. PLoS One, 8(8), e69265.
Horner, J. R., & Goodwin, M. B. (2006). Major cranial changes during Triceratops ontogeny. Proceedings of the Royal Society of London B: Biological Sciences, 273:2757-2761.
Makovicky, P.J. 2010. A redescription of the Montanoceratops cerorhynchus holotype with a review of referred material; pp. 68–82 in M.J. Ryan, B.J. Chinnery-Allgeier and D.A. Eberth (eds.) New perspectives on horned dinosaurs: The Royal Tyrell Museum Ceratopsian Symposium. Bloomington, IN: Indiana University Press.
Morschhauser, E. M. (2012). The anatomy and phylogeny of Auroraceratops (Ornithischia: Ceratopsia) from the Yujingzi Basin of Gansu Province, China. University of Pennsylvania.
Sereno, P.C. 2010. Taxonomy, cranial morphology, and relationships of parrot-beaked dinosaurs (Ceratopsia: Psittacosaurus). Pp. 21-58 in M.J. Ryan, B.J. Chinnery-Allgeier, and D.A. Eberth (eds.). New Perspectives on Horned Dinosaurs: the Royal Tyrrell Ceratopsian Symposium. Indiana University Press, 624p.
Xu, X., P.J. Makovicky, X.-L. Wang, M.A. Norell and H.-L. You. 2002. A ceratopsian dinosaur from China and the early evolution of Ceratopsia. Nature 416: 314–317.
You, H. L., Tanoue, K., & Dodson, P. (2008). New data on cranial anatomy of the ceratopsian dinosaur Psittacosaurus major. Acta Palaeontologica Polonica, 53(2), 183-196.

---

## Round 0.2 · Minor Revisions

Dear Claire,

As per PeerJ policies (https://peerj.com/about/policies-and-procedures/#data-materials-sharing), all the raw data (CT scan data in this case) have to be made available in a permanent public repository prior to formal acceptance. You indicate in your rebuttal letter that "the original scan data is available at any time by application to the Collection Manager at IVPP, Beijing", but you do not provide specific instructions for access in the manuscript. Please, address this issue.

Best regards,
Fabien

·

Basic reporting

See general comments

Experimental design

See general comments

Validity of the findings

See general comments

Additional comments

I find the already well-written manuscript much improved after the changes made by the authors. Most of the suggestions and comments of the reviewers were implemented, although it appears to be a bit selective for some suggestions. This however is a minor issue coming down to individual preferences and opinions, and should not preclude publication.
I am happy to see that the discussion has been revised considerably and provides a broader discussion of ontogenetic changes in ceratopsians and other dinosaurs. The list of morphological changes during ontogeny in the conclusions provides a useful summary New/updated figures are very helpful and support the text a lot better now.
All in all, I find the text and figures to be of a high standard and ready for publication.

---

## Round 0.3 · accepted · Accept

Dear Claire,

I accept your manuscript for publication in PeerJ. I request, however, that you provide in the "Materials and Methods" section a general contact for access to the data, i.e. one that will likely be valid in the distant future, when the current Collection Manager leaves IVPP (i.e. not drsmith@institution.xy but something like curator@institution.xy). You can make this edit while your manuscript is in Production (and thank the reviewers as well if you feel like it).

Thank you for submitting this interesting work to PeerJ!

Best regards,
Fabien